# Unlocking the Transferability of Tokens in Deep Models for Tabular Data

## Abstract

Fine-tuning a pre-trained deep neural network has become a successful paradigm in various machine learning tasks. However, such a paradigm becomes particularly challenging with tabular data when there are discrepancies between the feature sets of pre-trained models and the target tasks. In this paper, we propose TABTOKEN, a method aims at enhancing the quality of feature tokens (*i.e.*, embeddings of tabular features). TABTOKEN allows for the utilization of pre-trained models when the upstream and downstream tasks share overlapping features, facilitating model fine-tuning even with limited training examples. Specifically, we introduce a contrastive objective that regularizes the tokens, capturing the semantics within and across features. During the pre-training stage, the tokens are learned jointly with top-layer deep models such as transformer. In the downstream task, tokens of the shared features are kept fixed while TABTOKEN efficiently fine-tunes the remaining parts of the model. TABTOKEN not only enables knowledge transfer from a pre-trained model to tasks with heterogeneous features, but also enhances the discriminative ability of deep tabular models in standard classification and regression tasks.

## 1 Introduction

Deep learning has achieved remarkable success in various domains, including computer vision (Voulodimos et al., 2018), and natural language processing (Otter et al., 2020). While these fields have benefited greatly from deep learning, the application of deep models to tabular data is difficult (Gorishniy et al., 2021; Grinsztajn et al., 2022). Highly structured, tabular data is organized with rows representing individual examples and columns corresponding to specific features. Within domains such as finance (Addo et al., 2018), healthcare (Hassan et al., 2020), and e-commerce (Nederstigt et al., 2014), tabular data emerges as a common format where classical machine learning methods like XGBoost (Chen & Guestrin, 2016) have showcased strong performance. In recent years, deep models have been extended to tabular data, leveraging the ability of deep neural networks to capture complex feature interactions and achieve competitive results compared to boosting methods in certain tasks (Cheng et al., 2016; Guo et al., 2017; Popov et al., 2020; Arik & Pfister, 2021; Gorishniy et al., 2021; Katzir et al., 2021; Chang et al., 2022; Chen et al., 2022; Hollmann et al., 2023).

The "pre-training then fine-tuning" paradigm is widely adopted in deep learning. By reusing the pre-trained feature extractor, the entire model is subsequently fine-tuned on target datasets (Gando et al., 2016; Yang et al., 2017; Tan et al., 2018; Too et al., 2019; Subramanian et al., 2022). However, when it comes to tabular data, the transferability of pre-trained deep models faces unique challenges. The tabular features often possess semantic meanings and the model attempts to comprehend the relationships between features and targets. Each tabular feature has strong correspondence with model parameters, making it hard to directly transfer pre-trained deep learning models when encountering unseen features (Wang & Sun, 2022; Dinh et al., 2022; Hegselmann et al., 2023; Onishi et al., 2023; Zhu et al., 2023). For instance, different branches of financial institutions may share certain features when predicting credit card fraud, but each branch may also possess unique features associated with their transaction histories. Besides, the scarcity of available samples for fine-tuning on new datasets further complicates the knowledge transfer process.

In this paper, we focus on a crucial component of tabular deep models — the feature tokenizer, which is an embedding layer that transforms numerical or categorical features into high-dimensional vectors (tokens). The original features correspond specifically with these tokens, which are leveraged by

the top-layer models like transformers (Vaswani et al., 2017) and Multi-Layer Perceptrons to extract relationships between features (Gorishniy et al., 2021; 2022). Through the feature tokenizer, all features are transformed in the same manner, the tokens seem to be a tool to bridge two heterogeneous feature sets by reusing *tokens of shared features in a pre-trained model*. By transferring the feature tokens, the model achieves a reduction in the size of learnable parameters. The model may leverage knowledge acquired from pre-training data and enhance its generalization ability on the target dataset.

However, our observation reveals that the learned tokens exhibit random behavior and lack discriminative properties. Since learning the feature relationships is crucial as it enables the model to gain a deeper understanding of the underlying patterns within the data, the top-layer models encounter more difficulties in effectively learning from the semantically deficient tokens. As illustrated in Figure 1, if the feature tokens corresponding to the six possible values of "skin color" are randomly distributed, the decision boundary becomes complex. The correlation between skin color and ripeness needs to be learned by a more complex top-layer model. However, with a discriminative semantic distribution, a simple classifier can achieve accurate predictions. At this point, the tokens will contain potentially transferable knowledge, which may unlock the transferability of feature tokens.

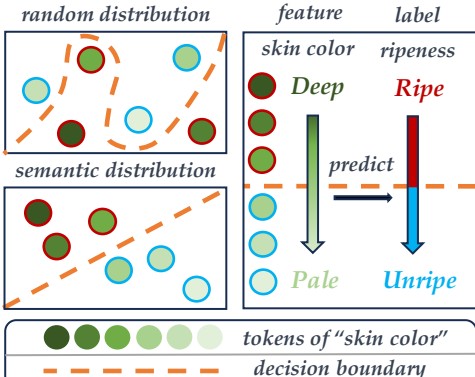

Figure 1: The toy example involves predicting ripeness based on the color of the watermelon, where ripe ones have a deep color and unripe ones are pale. Semantic distributions are more discriminative and possess the potential for transferability.

To address this issue, we propose TABTOKEN to introduce semantics to feature tokens, improving the transferability of feature tokenizers. Specifically, TABTOKEN aims to leverage the semantic information provided by the instance labels to understand feature relationships. Firstly, we represent an instance by averaging the tokens associated with all features. Based on the instance tokens, we introduce a contrastive token regularization objective that minimizes the distance between instances and their respective class centers. By incorporating this regularization into the pre-training stages, TABTOKEN enables the previously randomly distributed feature tokens to reflect actual semantic information, bolstering the model's transferability. Furthermore, the fine-tuning stage benefits from the enhanced pre-trained feature tokens, learning new modules under the constraint of regularization.

Our experiments demonstrate the efficacy of TABTOKEN in achieving model transfer across datasets with heterogeneous feature sets. TABTOKEN also improves the performance of deep models in standard tabular tasks involving classification and regression. The contributions of our paper are:

- We emphasize the significance of token quality in facilitating the transferability of tabular deep models when there is a change in the feature set between pre-training and downstream datasets. To the best of our knowledge, we are the first to focus on feature tokens in tabular transfer learning.
- We introduce TABTOKEN to enhance the transferability of deep tabular models by incorporating feature semantics into the tokens and enabling their utilization in downstream tasks.
- Through experiments on real-world datasets, we demonstrate the ability of TABTOKEN to reveal explainable feature semantics. Furthermore, our method showcases strong performance in heterogeneous transfer tasks as well as standard tabular tasks.

In the rest of this paper, we first describe the transfer problem and introduce our method. Subsequently, we present experiments and conclusions. The related work is reviewed in the Appendix.

## 2 RELATED WORK

We briefly review related approaches in deep tabular data learning and those transferring tabular models across feature spaces. More discussions are in Appendix A. Recently, a large number of deep learning models for tabular data have been developed (Cheng et al., 2016; Guo et al., 2017; Popov et al., 2020; Arik & Pfister, 2021; Katzir et al., 2021; Gorishniy et al., 2021; Chang et al., 2022; Chen et al., 2022; Hollmann et al., 2023), however, they do not possess the ability to transfer knowledge

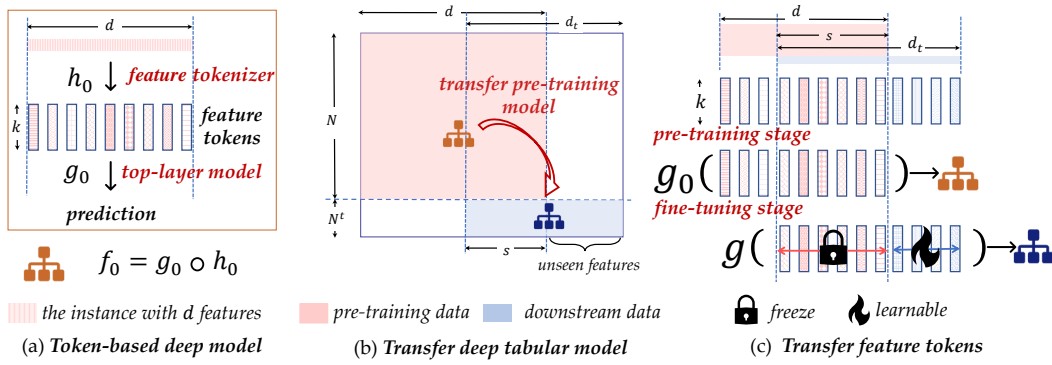

(a) *Token-based deep model*  (b) *Transfer deep tabular model*  (c) *Transfer feature tokens*

Figure 2: **Illustrations of the token-based model, transfer task, and the procedure of TABTOKEN. (a)** The token-based models $f_0$ for tabular data can be decomposed into a feature tokenizer $h_0$ and a top-layer model $g_0$. **(b)** In the transfer task, the downstream dataset consists of $s$ overlapping features with the pre-training dataset while also introducing $d_t - s$ unseen features. When the feature space changes, we expect to transfer the pre-trained model for downstream tasks. **(c)** In the pre-training stage, by employing token combination and regularization, TABTOKEN incorporates the semantics of labels into tokens. In the fine-tuning stage, TABTOKEN freezes the overlapping feature tokens of the pre-trained tokenizer, efficiently fine-tuning other modules.

from pre-trained tabular models across feature spaces. Instead of using feature descriptions as well as the language models (Wang & Sun, 2022; Dinh et al., 2022; Hegselmann et al., 2023), pseudo-Feature method (Levin et al., 2023), TabRet (Onishi et al., 2023), XTab (Zhu et al., 2023), and ORCA (Shen et al., 2023) improve the transferability of the model from via pseudo-feature or learned transformers. In this paper, we discover the untapped potential in improving the feature tokens and aim to develop a tokenizer with stronger transferability. In contrast to prior work, our emphasis lies in enhancing the quality of feature tokens and unlocking their transferability.

## 3  PRELIMINARY AND BACKGROUND

In this section, we introduce the task learning with tabular data, as well as the vanilla deep tabular models based on feature tokens. We also describe the scenario of transferring a pre-trained model to downstream tasks with heterogeneous feature sets.

### 3.1  LEARNING WITH TABULAR DATA

Given a labeled tabular dataset $\mathcal{D} = \{(\boldsymbol{x}_{i,:}, y_i)\}_{i=1}^{N}$ with $N$ examples (rows in the table). An instance $\boldsymbol{x}_{i,:}$ is associated with a label $y_i$. We consider three types of tasks: binary classification $y_i \in \{0, 1\}$, multiclass classification $y_i \in [C] = \{1, \ldots, C\}$, and regression $y_i \in \mathbb{R}$. There are $d$ features (columns) for an instance $\boldsymbol{x}_{i,:}$, we denote the $j$-th feature in tabular dataset as $\boldsymbol{x}_{:,j}$. The tabular features include numerical (continuous) type $\boldsymbol{x}_{:,j}^{\text{num}}$ and categorical (discrete) type $\boldsymbol{x}_{:,j}^{\text{cat}}$. Denote the $j$-th feature value of $\boldsymbol{x}_{i,:}$ as $x_{i,j}$. We assume $x_{i,j}^{\text{num}} \in \mathbb{R}$ when it is numerical (*e.g.*, the salary of a person). While for a categorical feature (*e.g.*, the occupation of a person) with $K_j$ discrete choices, we present $\boldsymbol{x}_{i,j}^{\text{cat}} \in \{0, 1\}^{K_j}$ in a one-hot form, where a single element is assigned a value of 1 to indicate the categorical value. We learn a model $f_0$ on $\mathcal{D}$ that maps $\boldsymbol{x}_{i,:}$ to its label $y_i$, and a generalizable model could extend its ability to unseen instances sampled from the same distribution as $\mathcal{D}$.

### 3.2  TOKEN-BASED METHODS FOR TABULAR DATA

There are several classical methods when learning on tabular data, such as logistic regression (LR) and XGBoost. Similar to the basic feature extractor when applying deep models over images and texts, one direct way to apply deep learning models on the tabular data is to implement model $f_0$ with Multi-Layer Perceptron (MLP) or Residual Network (ResNet). While for token-based deep learning models, the model $f_0$ is implemented by the combination of feature tokenizer $h_0$ and top-layer deep models $g_0$. The prediction for instance $\boldsymbol{x}_{i,:}$ is denoted as $f_0(\boldsymbol{x}_{i,:}) = g_0 \circ h_0(\boldsymbol{x}_{i,:}) = g_0(h_0(\boldsymbol{x}_{i,:}))$. We minimize the following objective to obtain the feature tokenizer $h_0$ and top-layer model $g_0$:

$$\min_{f_0 = g_0 \circ h_0} \sum_{i=1}^{N} \left[ \ell \left( g_0 \circ h_0(\boldsymbol{x}_{i,:}), y_i \right) \right], \tag{1}$$

where the loss function $\ell(\cdot, \cdot)$ measures the discrepancy between the prediction and the label. Feature tokenizer $h_0$ performs a similar role to the feature extractor in traditional image models Goodfellow et al. (2016); Gorishniy et al. (2021). It maps the features to high-dimensional tokens in a hidden space, facilitating the learning process of top-layer $g_0$. The top-layer model $g_0$ infers feature relationships based on the feature tokenizer $h_0$ and then performs subsequent classification or regression tasks.

**Feature Tokenizer** $h_0$ transforms the input $\boldsymbol{x}_i$ into a $d \times k$ matrix $\{\hat{\boldsymbol{x}}_{i,j}\}_{j=1}^{d}$. In detail, for a numerical feature value $x_{i,j}^{\mathrm{num}} \in \mathbb{R}$, it is transformed as $\hat{\boldsymbol{x}}_{i,j} = x_{i,j}^{\mathrm{num}} \cdot \boldsymbol{E}_j^{\mathrm{num}}$, where $\boldsymbol{E}_j^{\mathrm{num}}$ is a *learnable* $k$-dimensional vector (token) for the $j$-th feature. For a categorical feature $\boldsymbol{x}_{i,j}^{\mathrm{cat}} \in \{0, 1\}^{K_j}$, where there are $K_j$ discrete choices, the tokenizer is implemented as a lookup table. Given $\boldsymbol{E}_j^{\mathrm{cat}} \in \mathbb{R}^{K_j \times k}$ as the set of $K_j$ tokens, the tokenizer transform $\boldsymbol{x}_{i,j}^{\mathrm{cat}}$ as $\hat{\boldsymbol{x}}_{i,j} = \boldsymbol{x}_{i,j}^{\mathrm{cat}\top} \boldsymbol{E}_j^{\mathrm{cat}}$. Overall, the feature tokenizer maps different types of features to a unified form — $\boldsymbol{x}_{i,:}$ is transformed into a $d \times k$ matrix $\{\hat{\boldsymbol{x}}_{i,j}\}_{j=1}^{d}$ with a set of $k$-dimensional tokens. The model learns feature tokens $\boldsymbol{E}^{\mathrm{num}}$ or $\boldsymbol{E}^{\mathrm{cat}}$ during the training process. The feature tokenizer $h_0$ is a weighted or selective representation of feature tokens.

**Deep Top-layer Models**. After the tokenizer, the transformed tokens $h_0(\boldsymbol{x}_{i,:}) = \{\hat{\boldsymbol{x}}_{i,j}\}_{j=1}^{d}$ for instance $\boldsymbol{x}_{i,:}$ are feed into top-layer models. There can be various top-layer models, such as MLP, ResNet, and transformer. When the top-layer model is MLP or ResNet, it cannot directly process the set of tokens $h_0(\boldsymbol{x}_{i,:})$. The common approach is to concatenate $d$ tokens into a long $dk$-dimensional vector $\boldsymbol{T}_i^{\mathrm{con}} \in \mathbb{R}^{dk}$. Then, $\boldsymbol{T}_i^{\mathrm{con}}$ feeds into the top-layer model $g_0$:

$$f_0(\boldsymbol{x}_{i,:}) = g_0(\boldsymbol{T}_i^{\mathrm{con}}), \ \boldsymbol{T}_i^{\mathrm{con}} = \mathrm{concat}(h_0(\boldsymbol{x}_{i,:})) \ . \tag{2}$$

For the transformer, the token set of $\boldsymbol{x}_{i,:}$ directly feeds into the transformer layer without the need for concatenating. The input $h_0(\boldsymbol{x}_{i,:})$ and output $\boldsymbol{F}_i$ of transformer are both a set of $k$-dimension tokens $\boldsymbol{F}_i = \mathrm{TransformerLayer}(\dots(\mathrm{TransformerLayer}(h_0(\boldsymbol{x}_{i,:})))) \in \mathbb{R}^{d \times k}$, where the module $\mathrm{TransformerLayer}(\cdot)$ is described in Appendix C. Based on the transformed tokens, the results could be obtained based on a prediction head over the averaged token $f_0(\boldsymbol{x}_{i,:}) = \mathrm{Prediction}\left(\frac{1}{d}\sum_{j=1}^{d}\boldsymbol{F}_{i,j}\right)$, where $\boldsymbol{F}_{i,j}$ is the $j$-th token in the output token set $\boldsymbol{F}_i$ of transformer. The head $\mathrm{Prediction}(\cdot)$ consists of an activation function ReLU and a fully connected layer.

### 3.3 Transfer Learning with Overlapping and Unseen Features

Given another related downstream dataset $\mathcal{D}^t = \{(\boldsymbol{x}_{i,:}^t, y_i^t)\}_{i=1}^{N^t}$ with $N^t$ examples ($N^t \ll N$) and $d_t$ features, we aim to construct a new model $f$ for $D^t$ borrowing the knowledge of well-trained $f_0$. We assume there exists heterogeneity between $\mathcal{D}$ and $\mathcal{D}^t$, in their feature or label sets. We mainly consider the scenario that $\mathcal{D}$ and $\mathcal{D}^t$ have the same label space but different feature spaces. We explore more complex scenarios in subsection E.3, such as transfer between non-overlapping label spaces. We denote the features ranging from the $j$-th to the $m$-th as $\{\boldsymbol{x}_{:,j:m}\}$. The last $s$ features $\{\boldsymbol{x}_{:,d-s+1:d}\}$ in the pre-trained dataset are shared features (overlapping features), corresponding to the first $s$ features $\{\boldsymbol{x}_{:,1:s}^t\}$ in the fine-tuning dataset. The unseen feature set in the downstream dataset is $\{\boldsymbol{x}_{:,s+1:d_t}^t\}$. We pre-train $h_0$ and $g_0$ together on $\mathcal{D}$, then we fine-tune $f = g \circ h$ on $\mathcal{D}^t$. Our goal (illustrated in Figure 2) is to train $f$ with strong generalization ability in downstream tasks.

Inspired by the idea of "pre-training then fine-tuning" in traditional deep learning, since feature tokenizer $h_0$ is pre-trained on sufficient examples, we expect to reuse its weights $\{\boldsymbol{E}_j^{\mathrm{num}}\}_{j=1}^{d}$ or $\{\boldsymbol{E}_j^{\mathrm{cat}}\}_{j=1}^{d}$ to construct a better fine-tuning feature tokenizer $h$. As each token from the tokenizer is correspond to a specific feature and there are overlapping features $\{\boldsymbol{x}_{:,d-s+1:d}\}$, it should be possible to reuse the token directly when the feature space changes. The knowledge acquired from the upstream task appears to be present in the pre-trained feature tokens.

However, our observation experiments (in subsection 5.1) show that the tokens generated during the vanilla pre-training process display stochastic patterns. These feature tokens lack sufficient semantic information, hindering their ability to reflect discriminative feature relationships, thereby limiting their effectiveness in aiding top-layer models and transferability in transfer learning.

# 4 Improve Feature Tokens for Transferable Deep Tabular Models

To address the issue that the feature tokens obtained through vanilla training are distributed in an almost random manner, we analyze the significance of semantics in unlocking feature tokens' transferability, and propose a pre-training then fine-tuning procedure for transfer tasks.

## 4.1 Token Semantics Play a Role in Transferability

Although feature tokenizer replaces a sparse feature (*e.g.*, the one-hot coding of a categorical feature) with a dense high-dimensional vector, we provide an explanation in Appendix B that *feature tokenizer does not increase the model capacity*. Since the feature tokens are in correspondence with features, tokens should reflect the nature of features, implying that tokens may carry semantics. However, based on our analysis in Appendix A, it is evident that the feature tokenizer does not increase the model capacity. During vanilla training, different tokens are concatenated and then predicted by different parts of the classifier. Considering an extreme scenario, the model could accurately predict an instance even when feature tokens are generated randomly, by only optimizing the classifier. The feature token itself is not directly used for prediction. Consequently, as long as the top-layer model is sufficiently strong for the current task, the trained tokens do not retain enough knowledge for transfer.

Tokens that lack semantics or even resemble randomness lack discriminative power and hold little value for reuse. Unlike in the image or text domains, where the feature extractors capture the semantic meaning with their strong capacity and enable model transferability across tasks (Simonyan & Zisserman, 2015; Devlin et al., 2018), unlocking the transferability of feature tokenizer in the tabular model is challenging. Without additional information, it is not possible to directly learn semantically meaningful tokens. The downstream task also fails to transfer the learned knowledge in overall model due to the changes of feature space. The correspondence between tokens and features may enable token-based deep models to be transferable for new tasks involving shared features and unseen features, so it is crucial to imbue feature tokens with semantics.

## 4.2 Semantic Enriched Pre-Training

The label assigned to an instance could be additional information that help feature tokens understand semantics. For example, for the feature "occupation" with values like "entrepreneur", "manager", and "unemployed", if the label indicates whether the individual will invest in financial products, typically, the labels for "entrepreneur" and "unemployed" would be "yes", while "unemployed" might be associated with low probability. By grouping instances with the same label together, semantically similar tokens corresponding to "entrepreneur" and "manager" would cluster. Therefore, to achieve the goal of imbuing tokens with semantics, we utilize instance labels as additional supervision.

To facilitate direct distance computation for grouping instances, we need to represent each instance by one token instead of a token set. Token combination transforms the set of tokens $h_0(\boldsymbol{x}_{i,:}) = \{\hat{\boldsymbol{x}}_{i,j}\}_{j=1}^d$ into an instance token $\boldsymbol{T}_i$ of $\boldsymbol{x}_{i,:}$. As in Equation 2, the usual combination operation is concatenating, which makes $\boldsymbol{T}_i^{\text{con}}$ a high-dimensional vector. However, when the order of input features is permuted, applying concatenating to the same $\boldsymbol{x}_{i,:}$ after the feature tokenizer will result in different $\boldsymbol{T}_i^{\text{con}}$. Besides, the vector values at the same position in different feature tokens will correspond to different weights in the subsequent predictor. This implies that, for different features $\boldsymbol{x}_{:,m}$ and $\boldsymbol{x}_{:,n}(m \neq n)$, the meanings of $\hat{\boldsymbol{x}}_{i,m} \in \mathbb{R}^k$ and $\hat{\boldsymbol{x}}_{i,n} \in \mathbb{R}^k$ are not the same in the corresponding dimension. Therefore, concatenating fails to consider *the alignment between features*.

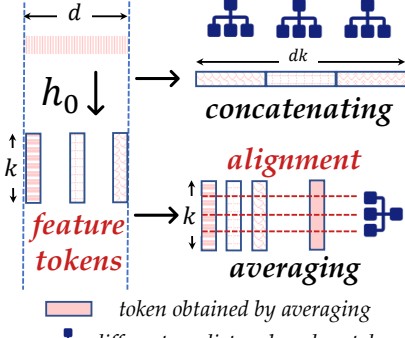

Figure 3: Token averaging outperforms combination as it aligns distinct feature tokens, which enhances the semantics.

Instead, we propose utilizing averaging for token combination, which averages the vector values at the same position in different feature tokens. Averaging also makes instance tokens remain unchanged regardless of the order of input features. We obtain the representation $\boldsymbol{T}_i^{\text{avg}} \in \mathbb{R}^k$ for instance $\boldsymbol{x}_i$ in the following form: $\boldsymbol{T}_i^{\text{avg}} = \frac{1}{d} \sum_{j=1}^d \hat{\boldsymbol{x}}_{i,j}$. For notational clarity, we drop the superscript avg of $\boldsymbol{T}_i^{\text{avg}}$. We

calculate the the class center $S_{y_i} \in \mathbb{R}^k$ for $x_{i,:}$ by averaging instance tokens belong to the same class: $S_{y_i} = \frac{1}{N^{y_i}} \sum_{y_p=y_i} T_p$, where $\{T_p\}_{y_p=y_i}$ are the $N^{y_i}$ instance tokens with the same class to $x_{i,:}$. $S_{y_i}$ is the class center belongs to the label $y_i$ of $x_{i,:}$. We anticipate that instance tokens belonging to the same class will exhibit proximity. As instance tokens are derived from averaging the feature tokens, the tokens that share similar semantic relevance to the class should cluster together. Therefore, we pull instance tokens toward their class center via a contrastive regularization $\Omega : \mathbb{R}^{N \times k} \to \mathbb{R}$:

$$\Omega\left(\{T_i\}_{i=1}^N\right) = \frac{1}{N} \sum_{i=1}^N \|T_i - S_{y_i}\|_2^2 \,.$$

We name this contrastive regularization as "**Contrastive Token Regularization (CTR)**". For token regularization, we calculate the regularization term on instance tokens and add the term to training loss $\ell$. The objective is to minimize the following loss:

$$\min_{f_0=g_0 \circ h_0} \sum_{i=1}^N \left[\ell(g_0 \circ h_0(x_{i,:}), y_i)\right] + \beta \Omega\left(\{T_i\}_{i=1}^N\right), \tag{3}$$

where $\beta$ is a hyperparameter that controls the regularization term. In the pre-training stage, we learn the feature tokenizer $h_0$ and the top-layer model $g_0$ simultaneously. Different from the objective in Equation 1, the feature tokens $\{E_j^{\text{num}}\}_{j=1}^d$ or $\{E_j^{\text{cat}}\}_{j=1}^d$ are subject to two constraints. Firstly, they need to be adjusted in a way that enhances the predictions of the top-layer model. Secondly, a regularization is employed to ensure that the tokens retain their semantic.

In practice, only the samples within each batch are used to compute the center. For regression tasks, the regularization term is computed by dividing the target values into two pseudo-classes using the median as a threshold. For instance, if the median of the target values in the dataset is 0.5, samples with values greater than 0.5 are labeled as class 1, while less than 0.5 are labeled as class 2. These pseudo-labels are only utilized for computing the CTR. In Appendix E, we compare the effect of CTR with other contrastive losses, verifying that our simple regularization has obvious advantages.

### 4.3 TOKEN REUSED FINE-TUNING

Through the utilization of the CTR, we can enhance the feature tokenizer during the pre-training stage. In this subsection, we elucidate the reuse process based on high-quality tokens. We pre-train $g_0 \circ h_0$ with CTR, which is essential for the transferability of feature tokens. For the ease of expression, we use $\{E_j^{\text{pre}}\}_{j=1}^d$ to represent $\{E_j^{\text{num}}\}_{j=1}^d$ or $\{E_j^{\text{cat}}\}_{j=1}^d$, which are the feature tokens of the pre-training feature tokenizer $h_0$. In the transfer task, we expect to reuse $\{E_j^{\text{pre}}\}_{j=1}^d$ to construct fine-tuning feature tokenizer $h$, then we fine-tuning $g \circ h$ on downstream dataset $\mathcal{D}^t$.

There are $d$ pre-training features $\{x_{:,1:d}\}$ and $d_t$ downstream features $\{x_{:,1:d_t}^t\}$. As illustrated in Figure 2(b), in the downstream task, there are two feature sets: $s$ overlapping features $\{x_{:,1:s}^t\}(\{x_{:,d-s+1:d}\})$ and $d_t - s$ unseen features $\{x_{:,s+1:d_t}^t\}$. To construct the fine-tuning tokenizer $h$, for overlapping features, $h$ fix the pre-trained feature tokens $\{E_j^{\text{pre}}\}_{j=d-s+1}^d$ of $h_0$ and transfer. For unseen features, $h$ firstly initializes the remaining feature tokens based on the averaging of all pre-trained feature tokens: $\frac{1}{d} \sum_{j=1}^d E_j^{\text{pre}} \in \mathbb{R}^k$. Then $h$ fine-tunes these learnable tokens.

After building $h$, we freeze the overlapping feature tokens. The learnable modules are top-layer model $g$ (transformer) and unseen feature tokens in $h$. To regularize feature tokens for unseen features, we utilize averaging and CTR to perform fine-tuning, too. Continuing with the notations from Equation 3, we optimize model $f = g \circ h$ via minimizing the following objective on downstream dataset $\mathcal{D}^t$:

$$\min_{f=g \circ h} \quad \sum_{i=1}^{N^t} \left[\ell(g \circ h(x_{i,:}^t), y_i^t)\right] + \beta \cdot \frac{1}{N^t} \sum_{i=1}^{N^t} \left\|T_i - S_{y_i^t}\right\|_2^2$$

$$\text{s.t.} \quad h(x_{i,:}^t) = H(x_{i,:}^t), \ H(x_{i,j}^t) = \begin{cases} h_0(x_{i,j}^t), & 1 \le j \le s, \\ \hat{x}_{i,j}^t, & s+1 \le j \le d_t, \end{cases}$$

where $h_0$ remains fixed and $\beta$ is a pre-defined coefficient for adjusting the regularization term.

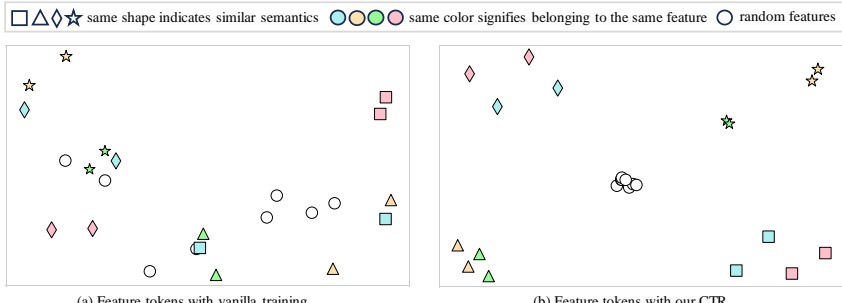

(a) Feature tokens with vanilla training.    (b) Feature tokens with our CTR.

Figure 4: **Feature tokens on synthetic dataset**. Colors indicate which feature the tokens belong to, and the same shapes indicate semantically similar tokens. Colorless circles represent tokens of random features. **(a)**: Categories with similar semantics across different features are not captured in the tokens. Tokens from random features may come close to other tokens that are relevant to the target, thereby influencing the prediction. **(b)**: Tokens with similar semantics exhibit a clear clustering phenomenon, while tokens representing random features are tightly clustered together in the center.

**Summary of TABTOKEN**. When there is a change in the feature space, we aim to transfer the tokens corresponding to shared features. In the pre-training stage, we enhance the quality of tokens by introducing semantics, making them transferable. In the fine-tuning stage, we continue to constrain new feature tokens using both the frozen pre-trained feature tokens and CTR. TABTOKEN unlocks the tokens' transferability through a well-designed combination of pre-training and fine-tuning processes.

## 5 EXPERIMENTS

In this section, we demonstrate the performance of the feature tokens obtained by TABTOKEN in transfer tasks using real-world tabular data. The implementation details are introduced in Appendix F.

### 5.1 VISUALIZATION ON TOKEN SEMANTICS

Different from visualizations on the penultimate layer's embeddings (Huang et al., 2020), the feature tokens are feature-specific embeddings near the input of the tabular model, which better reveals the quality of the feature tokenizer.

**Semantical Relationships between Different Features**. We construct a synthetic four-class classification dataset consisting of 6 features, each feature has 4 categorical values. Thus, we get $6 \times 4 = 24$ feature tokens in TABTOKEN. We first construct highly correlated features as follows: when we generate an instance, a pair of features have a one-to-one correspondence on the categorical choices. The relationship between the features and labels is closely tied to the feature values. As a result, these semantically similar tokens come from different features. We expect these tokens to be close to each other. Besides, we construct noise features with random values. We expect random feature tokens to be far from those that contribute to predictions. Otherwise, random feature tokens will exert similar influences on the subsequent top-layer model's prediction process.

We train a one-layer Linear model upon a 2-dimensional feature tokenizer. The results are in Figure 4. We find feature tokens generated by vanilla training exhibit a random distribution without discernible patterns. When training with CTR, all groups of tokens are separately clustered together. TABTOKEN enables the trained feature tokens to capture the semantic correlation between different features. Besides, some random tokens (the colorless circles) are closely located to those of semantically meaningful features with vanilla training. However, they occupy the same region in the latent space, staying away from other feature tokens, once with CTR. The phenomenon validates that CTR reduces the distance between instance tokens of the same class, which helps *identify noise features*.

**Category Relationships within Single Feature**. We selected three categorical features, "job", "education", and "marital", from the real-world dataset "bank-marketing", whose target is to predict the success rate of a customer purchasing financial products. We aim for TABTOKEN to facilitate the learning of semantics and help feature tokens capture the category relationships within a single feature. We train a three-layer transformer on a 64-dimensional feature tokenizer.

Figure 5 shows the learned feature tokens, which are consistent with the arrangements within the features, such as "tertiary" → "secondary" → "primary" in feature "education". Besides, the tokens

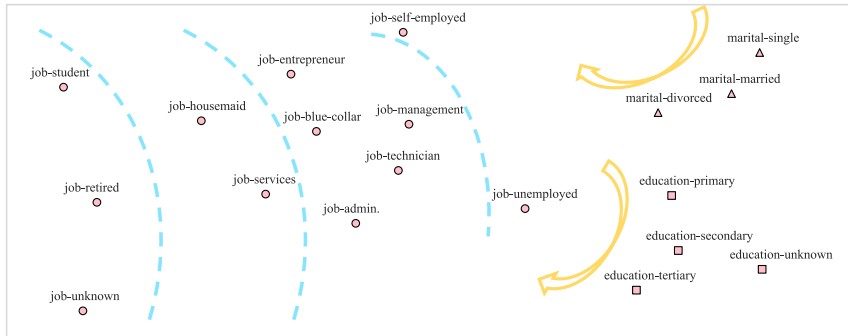

Figure 5: **Feature tokens trained with CTR on bank-marketing dataset**. The tokens of feature "job" depict a distribution based on job types. The hierarchical pattern is in relation to the probability associated with purchasing financial products. The distribution of tokens for the feature "education" and feature "marital" aligns perfectly with their respective semantic order. Tokens with vanilla training are shown in Figure 9.

Table 1: Results for 5-shot downstream tasks. TABTOKEN outperforms other baselines in the transfer setting, reflecting the transferable nature of the tokens obtained by CTR. †: TabPFN does not support regression tasks. (↑ ~ accuracy, ↓ ~ RMSE). The whole results with the standard deviation are listed in Table 7.

| | Eye↑ | Colon↑ | Clave↑ | Cardio↑ | Jannis↑ | Htru↑ | Breast↑ | Elevators↓ | Super↓ | Volume↓ |
|---|---|---|---|---|---|---|---|---|---|---|
| SVM | 0.3621 | 0.5921 | 0.3482 | 0.6036 | 0.3512 | 0.8376 | 0.8211 | 0.0088 | 33.9036 | 124.2391 |
| XGBoost | 0.3699 | 0.5676 | 0.3506 | 0.5703 | 0.3222 | 0.8369 | **0.8453** | 0.0090 | 34.6605 | 123.9724 |
| FT-trans | 0.3916 | 0.5792 | 0.3584 | 0.6064 | 0.3064 | 0.8252 | 0.8275 | 0.0081 | 31.3274 | 122.8319 |
| TabPFN† | 0.3918 | 0.5809 | 0.3733 | 0.5965 | 0.3601 | 0.8371 | 0.7438 | - | - | - |
| SCARF | 0.3371 | 0.6048 | 0.2144 | 0.5547 | 0.3523 | 0.8131 | 0.7063 | 0.0097 | 39.9343 | 124.5373 |
| TabRet | 0.3477 | 0.4688 | 0.2393 | 0.4329 | 0.3545 | 0.8231 | 0.7252 | 0.0094 | 41.2537 | 126.4713 |
| XTab | 0.3851 | 0.5964 | 0.3627 | 0.5856 | 0.2868 | 0.8363 | 0.8145 | 0.0077 | 38.5030 | 119.6656 |
| ORCA | 0.3823 | 0.5876 | 0.3689 | 0.6042 | 0.3413 | 0.8421 | 0.8242 | 0.0082 | 37.9436 | 121.4952 |
| TABTOKEN | **0.3982** | **0.6074** | **0.3741** | **0.6229** | **0.3687** | **0.8459** | 0.8284 | **0.0074** | **30.9636** | **118.7280** |

associated with the feature "job" exhibit a hierarchical pattern, where certain jobs like "retired", "student", and "unknown" form one layer, while jobs such as "housemaid" and "service" form another layer. Since individuals with these high-paying jobs are highly likely to successfully purchase financial products, those tokens align with their semantics when predicting the target of the dataset.

## 5.2 TOKEN MATTERS IN TRANSFER

We first evaluate TABTOKEN in transfer tasks on classification and regression. Further investigations are based on different shot numbers, overlapping ratios, and pre-training top-layer models.

**Datasets**. We conduct experiments on 10 open-source tabular datasets from various fields, including medical, physical and traffic domains. The datasets we utilized encompass both classification and regression tasks, as well as numerical and categorical features (details in Table 3).

**Experimental setups**. To explore transfer challenges in the presence of feature overlapping, we follow the data split process in (Nguyen et al., 2012; Hou & Zhou, 2018; Beyazit et al., 2019; Ye et al., 2021). We split the tabular dataset into pre-training dataset and fine-tuning dataset. The detailed split property and evaluation process for each datasets are shown in Table 4 and subsection D.3. From the downstream dataset, we randomly sample subsets as few-shot tasks. For instance, the dataset Eye has 3 classes, we split 5 samples from each class, obtaining a 5-shot downstream dataset with $3 \times 5$ samples. For regression tasks, each 5-shot downstream dataset consists of 5 samples. We report the average of 300 (30 subsets × 10 random seeds) results. In the fine-tuning stage, we are not allowed to utilize the validation set and pre-training set due to the constraints of limited data.

**Baselines**. We use four types of baselines: (1) The methods that train models from scratch on downstream datasets. We choose Support Vector Machine (SVM), XGBoost (Chen & Guestrin, 2016), and FT-trans (Gorishniy et al., 2021) which have strong performance on tabular data. TabPFN (Hollmann et al., 2023) specifically designed for small classification datasets is also compared. (2) The methods that use contrastive learning for pre-training: SCARF (Bahri et al., 2022). (3) Pre-training and fine-tuning methods designed specifically for tabular data with overlapping features: TabRet (Onishi

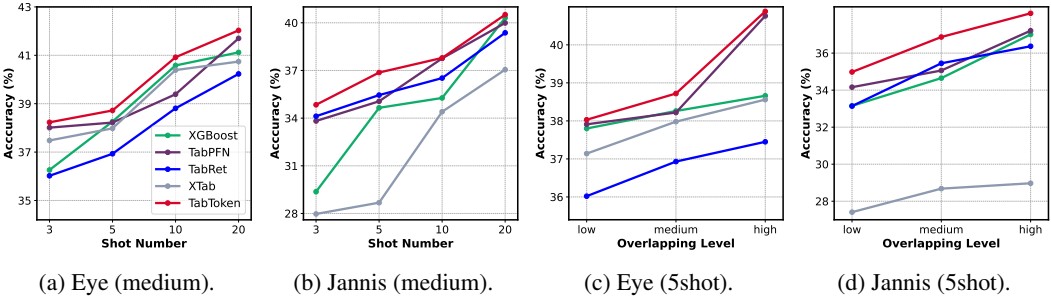

(a) Eye (medium).          (b) Jannis (medium).          (c) Eye (5shot).          (d) Jannis (5shot).

Figure 6: Results for different shots and overlapping ratios. TABTOKEN outperforms other baselines with different degrees of data limitations. When the overlapping feature ratio improves, TABTOKEN can leverage a larger proportion of pre-trained feature tokens, leading to improved performance.

et al., 2023). (4) The methods that transfer large-scale pre-trained transformers for tabular data: XTab (Zhu et al., 2023) and ORCA (Shen et al., 2023). We compare more baselines in Appendix E.

**Results**. Table 1 shows the results of different methods in the downstream tasks. For non-transfer methods, our advantage lies in the reuse of high-quality tokens. Self-supervised transfer method SCARF struggles to adapt well to changes in the feature space. Although TabRet aims at transfer setting with overlapping features, it does not perform well on limited data. Both XTab, which leverages a significant amount of pre-trained data, and ORCA, which utilizes large-scale pre-trained models, are unable to surpass TABTOKEN. We present more results of "pretraining then finetuning" real-world applications with overlapping features in Table 2.

**Different Shots and Overlapping Ratios**. In order to fully verify the robustness of TABTOKEN, we conduct experiments on $\{3, 5, 10, 20\}$-shot and adjust the overlapping ratio to three levels $\{$low, medium, high$\}$. Figure 6 shows that TABTOKEN can maintain the transfer effect in different shots and overlapping ratios. The specific number of features for different overlapping ratios is in Table 5.

**Different Top-layer Model Types**. In the pre-training process, we combine the tokenizer with different top-layer models, and transfer the pre-trained feature tokens to a new transformer for the downstream task. We report the results of different pre-training strategies and pre-trained model types: $\{$MLP, ResNet, Transformer$\}$. Figure 7 shows that even when the pre-training and downstream model types are different , the feature tokens trained with CTR gain transferability. Averaging also plays a role in TABTOKEN. Therefore, token matters in transfer.

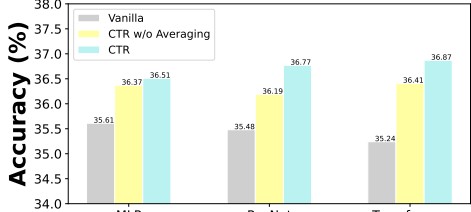

Figure 7: The results of 5-shot downstream tasks in Jannis. "Vanilla" means transferring feature tokens obtained by vanilla pre-training.

**Real-world medical applications with overlapping features**. We collect the Behavioral Risk Factor Surveillance System dataset in year 2015 for pre-training, which has 340057 samples. Two binary classification datasets, Diabetes and Stroke, are used for fine-tuning, which have 21 and 10 features, respectively. These two datasets have a feature overlapping ratio of approximately 50% with the pre-training dataset. Experimental results are listed in Table 2. TABTOKEN achieves the best results among others.

Table 2: The results of 5-shot classification for real-world cross-domain transfer.

| Dataset | Diabetes | Stroke |
|---------|----------|--------|
| XGBoost | 0.6252 | 0.5382 |
| FT-trans | 0.6126 | 0.5725 |
| SCARF | 0.6014 | 0.5697 |
| TabRet | 0.6259 | 0.5834 |
| TabToken | **0.6343** | **0.5845** |

## 6 CONCLUSION

Transferring a pre-trained tabular model effectively can enhance the learning of a downstream tabular model and improve its efficiency. We propose TABTOKEN to improve the transferability of deep tabular models by improving the quality of their crucial components — feature tokens. We introduce a contrastive objective that regularizes the tokens and incorporates feature semantics into tokens. Experimental results on diverse tabular datasets validate the effectiveness of our approach.

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

We highlight the significance of feature tokens when reusing a pre-trained deep tabular model from a task with an overlapping feature set. Our proposed method, TABTOKEN, focuses on improving the quality of feature tokens and demonstrates strong performance in both cross-feature-set and standard tabular data experiments. The Appendix consists of six sections:

- Appendix A: We introduce the related work of TABTOKEN.
- Appendix B: We demonstrate that the feature tokenizer does not increase the model capacity, which emphasizes the importance of enhancing the quality of tokens.
- Appendix C: We describe the architectures of several top-layer models in tabular deep learning, including MLP, ResNet, and Transformer.
- Appendix D: We provide details on generating synthetic datasets and real-world datasets. We specify how to split the dataset for evaluation.
- Appendix E: Additional experiments are presented, including comparisons with more variants and baselines. We extend our approach to more complex scenarios with different label spaces and non-overlapping features.
- Appendix F: Implementation details of baseline methods and TABTOKEN are provided.

## APPENDIX A  DETAILED DISCUSSIONS ON RELATED METHODS

**Standard Tabular Data Learning**. Tabular data is a prevalent data format in numerous real-world applications, spanning click-through rate prediction (Richardson et al., 2007), fraud detection (Awoyemi et al., 2017), and time-series forecasting (Ahmed et al., 2010). Traditional machine learning methods for tabular data mainly focus on feature engineering and hand-crafted feature selection. Popular algorithms encompass decision trees, random forests, and gradient boosting machines. Among them, XGBoost (Chen & Guestrin, 2016), LightGBM (Ke et al., 2017), and CatBoost (Prokhorenkova et al., 2018) are widely employed tree-based models that exhibit comparable performance for tabular data. Deep learning models have shown promise in automatically learning feature representations from raw data and have achieved competitive performance across diverse tabular data applications, such as click-through rate prediction (Zhang et al., 2021) and time-series forecasting (Lim & Zohren, 2021).

**Deep Tabular Data Learning**. Recently, a large number of deep learning models for tabular data have been developed (Cheng et al., 2016; Guo et al., 2017; Popov et al., 2020; Arik & Pfister, 2021; Katzir et al., 2021; Gorishniy et al., 2021; Chang et al., 2022; Chen et al., 2022; Hollmann et al., 2023). These models either imitate the tree structure or incorporate popular deep learning components. They have demonstrated superior performance compared to traditional machine learning methods, especially in sparse and high-dimensional feature spaces. However, deep learning models for tabular data struggle to learn high-order feature interactions (Grinsztajn et al., 2022), which are crucial in many tabular data applications, and require substantial amounts of training data. Boosting methods (Chen & Guestrin, 2016; Ke et al., 2017; Prokhorenkova et al., 2018) can effectively capture these interactions and are more robust to data limitations. While deep models may not surpass tree-based models entirely on tabular data, they offer greater flexibility and can be customized for complex and specific tasks, especially when faced with changing factors such as features and labels.

**Transferring Tabular Models across Feature Spaces**. Transferring knowledge from pre-trained models across feature spaces can be challenging but also beneficial, as it reduces the need for extensive data labeling and enables efficient knowledge reuse (Hou & Zhou, 2018; Zhang et al., 2020; Ye et al., 2021; Hou et al., 2021; 2022). In real-world applications such as healthcare, there are numerous medical diagnostic tables. These tables usually have some features in common such as blood type and blood pressure. For rare diseases with limited data, knowledge transfer from other diagnostic tables with overlapping features becomes beneficial. When feature space changes, language-based methods assume there are semantic relationships between the descriptions of features (Wang & Sun, 2022; Dinh et al., 2022; Hegselmann et al., 2023), and rely on large-scale language models.

However, sufficient textual descriptions are not always the case in tabular data. Without the requirement for language, Pseudo-Feature method (Levin et al., 2023) utilize pseudo-feature models individually for each new feature, which is computationally expensive in our broader feature space adaptation scenario. TabRet (Onishi et al., 2023) utilizes masked autoencoding to make transformer work in downstream tasks. XTab (Zhu et al., 2023) is dedicated to enhancing the transferability of the transformer (one type of the top-layer models), while we discover the untapped potential in improving the feature tokens and aim to develop a tokenizer with stronger transferability. To transfer pre-trained

large language models to tabular tasks, ORCA (Shen et al., 2023) trains an embedder to align the source and target distributions. Our approach, on the other hand, centers on how to directly transfer the embedder (tokenizer). In contrast to prior work, our emphasis lies in enhancing the quality of feature tokens and unlocking their transferability.

## APPENDIX B  THE TOKENIZER DOES NOT INCREASE THE MODEL CAPACITY

Unlike directly handling input data with a deep neural network, the token-based deep models add another embedding layer, which transforms the raw feature into a set of high-dimensional embeddings. In particular, given a labeled tabular dataset $\mathcal{D} = \{(\boldsymbol{x}_i, y_i)\}_{i=1}^N$ with $N$ instances (rows in the table), the feature tokenizer maps both categorical and numerical feature value $x_{i,j}$ to a $k$-dimensional vector. One of the main motivations of the feature tokenizer is to replace a sparse feature (*e.g.*, the one-hot coding of a categorical feature) with a dense high-dimensional vector with rich semantic meanings (Zhang et al., 2016; Guo et al., 2017; Huang et al., 2020; Liu et al., 2021; Zhao et al., 2021; Almagor & Hoshen, 2022). However, in this section, we analyze the feature tokenizer and show that they cannot increase the capacity of deep tabular models.

Following the same notation in the main text, we denote the $j$-th feature of the data as $\boldsymbol{x}_{:,j}$. Then we take the classification task as an example and analyze both numerical and categorical features.

**Numerical Features**. If the $j$-th feature is numerical, then the $j$-th element of an instance $\boldsymbol{x}_i$ is $x_{i,j} \in \mathbb{R}$. When we classify the label of $\boldsymbol{x}_i$ directly, we learn a classifier $w_0 \in \mathbb{R}$ for $x_{i,j}$, which predicts $x_{i,j}$ with $w_0^\top \cdot x_{i,j}$. While based on the feature tokenizer, we allocate a learnable embedding $\boldsymbol{E}_j \in \mathbb{R}^{1 \times k}$ for the feature $\boldsymbol{x}_{:,j}$, and transform $x_{i,j}$ with $\hat{\boldsymbol{x}}_{i,j} = x_{i,j} \cdot \boldsymbol{E}_j \in \mathbb{R}^{1 \times k}$. Based on the tokenized numerical feature $\hat{\boldsymbol{x}}_{i,j}$, the classifier becomes a vector $\boldsymbol{w} \in \mathbb{R}^k$, and the prediction works as follows:

$$f(\boldsymbol{x}_{i,j}) = \boldsymbol{w}^\top (x_{i,j} \cdot \boldsymbol{E}_j)^\top = (\boldsymbol{w}^\top \boldsymbol{E}_j^\top) \cdot x_{i,j} = w'^\top \cdot x_{i,j},$$

where $w'^\top = \boldsymbol{w}^\top \boldsymbol{E}_j^\top \in \mathbb{R}$. Therefore, it has the same effect as the original learning approach $w_0$.

**Categorical Features**. If the $j$-th feature is categorical with $K_j$ choices, we rewrite $\boldsymbol{x}_{i,j}$ in the one-hot coding form, *i.e.*, $\boldsymbol{x}_{i,j} \in \{0,1\}^{K_j} \in \mathbb{R}^{K_j}$. Assume the classifier for the one-hot feature is $\boldsymbol{\eta}_0 \in \mathbb{R}^{K_j}$, which predicts $\boldsymbol{x}_{i,j}$ with $\boldsymbol{\eta}_0^\top \boldsymbol{x}_{i,j}$. The tokenizer for a categorical feature works as a lookup table. Given $\boldsymbol{E}_j = \{\boldsymbol{e}_p\}_{p=1}^{K_j} \in \mathbb{R}^{K_j \times k}$ is the set of candidate tokens for the $j$-th feature, then we have $\hat{\boldsymbol{x}}_{i,j} = \boldsymbol{x}_{i,j}^\top \boldsymbol{E}_j \in \mathbb{R}^{1 \times k}$, which selects the corresponding token based on the index of the categorical feature value set. The classifier for the tokenized feature is $\boldsymbol{\eta} \in \mathbb{R}^k$ and the prediction is:

$$f(\boldsymbol{x}_{i,j}) = \boldsymbol{\eta}^\top (\boldsymbol{x}_{i,j}^\top \boldsymbol{E}_j)^\top = \boldsymbol{\eta}^\top \boldsymbol{E}_j^\top \boldsymbol{x}_{i,j} = \left[ \boldsymbol{\eta}^\top \boldsymbol{e}_1, \boldsymbol{\eta}^\top \boldsymbol{e}_2, \ldots, \boldsymbol{\eta}^\top \boldsymbol{e}_{K_j} \right] \boldsymbol{x}_{i,j} = \boldsymbol{\eta}'^\top \boldsymbol{x}_{i,j},$$

where $\boldsymbol{\eta}'^\top = \left[ \boldsymbol{\eta}^\top \boldsymbol{e}_1, \boldsymbol{\eta}^\top \boldsymbol{e}_2, \ldots, \boldsymbol{\eta}^\top \boldsymbol{e}_{K_j} \right] \in \mathbb{R}^{1 \times K_j}$. Therefore, the representation ability of the token-based model is the same as the original one $\boldsymbol{\eta}_0$.

**Summary**. The feature tokenizer does not increase the model capacity. Training with feature tokens cannot automatically associate the feature semantics with the tokens. Therefore, we incorporate the feature semantics into the tokens with a contrastive regularization in TABTOKEN, which makes the learned tokens facilitate the reuse of tabular deep models across feature sets.

## APPENDIX C  TOP-LAYER MODELS

Based on the transformed tokens, deep tabular models implement the classifier with various top-layer models, *e.g.*, MLP, ResNet, and Transformer. In this section, we formally describe the details of these models, whose architectures mainly follow the designs in (Gorishniy et al., 2021). However, instead of applying MLP and ResNet on raw features, we treat MLP and ResNet as the top-layer model upon the feature tokenizer. It is notable that the output of the feature tokenizer is a set of vectors. The main procedure is illustrated in Figure 8.

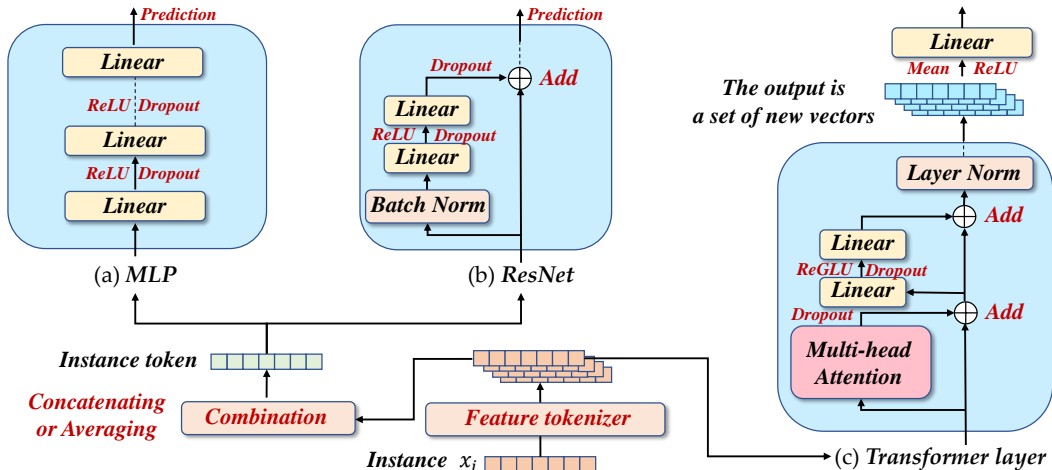

Figure 8: **The feature tokenizer and top-layer models**. (a) The Multi-Layer Perceptron needs the feature combination to process the input set of vectors. (b) The same to MLP, the Residual Network needs the feature combination to utilize a feature tokenizer. (c) A set of vectors can feed into the Transformer layer directly. The output is also a set of vectors, different from using a class token, we take the average of these vectors to obtain the vector used for the final prediction.

## C.1   MLP

Given a set of tokens, we transform them into a vector via token averaging or concatenating as described in the main text. With a bit of abuse of notation, we denote the processed token as $x$, whose dimension is $k$ or $dk$ when we average or concatenate the tokens, respectively. The MLP architecture is designed as follows based on $x$:

$$\text{MLP}(x) = \text{Linear}(\text{MLPBlock}(\dots(\text{MLPBlock}(x))),$$
$$\text{MLPBlock}(x) = \text{Dropout}(\text{ReLU}(\text{Linear}(x))),$$

where $\text{Linear}(\cdot)$ is a fully connected layer, which performs a linear transformation on the input data by applying a matrix multiplication followed by an optional bias addition. $\text{Dropout}(\cdot)$ is a regularization technique, which works by randomly deactivating a certain percentage of neurons during each training step. $\text{ReLU}(\cdot)$ is a simple and computationally efficient non-linear function that introduces non-linearity into the networks.

## C.2   RESNET

The same as MLP, we employ ResNet on the feature tokens, and the architecture is:

$$\text{ResNet}(x) = \text{Prediction}\left(\text{ResNetBlock}\left(\dots(\text{ResNetBlock}\left(\text{Linear}(x)\right))\right)\right),$$
$$\text{ResNetBlock}(x) = x + \text{Dropout}(\text{Linear}(\text{Dropout}(\text{ReLU}(\text{Linear}(\text{BatchNorm}(x))))))),$$
$$\text{Prediction}(x) = \text{Linear}\left(\text{ReLU}\left(\text{BatchNorm}\left(x\right)\right)\right),$$

where $\text{BatchNorm}(\cdot)$ is a technique used in neural networks during training (Ioffe & Szegedy, 2015). $\text{BatchNorm}(\cdot)$ normalizes the activations of a specific layer by adjusting and scaling them based on the statistics of the mini-batch.

## C.3   TRANSFORMER

Different from MLP or ResNet, Transformer processes the set of tokens simultaneously, and the output of Transformer is a set of tokens with the same form as the input one. Assume the input matrix $X$ is the set of $k$-dimensional vectors $X \in \mathbb{R}^{d \times k}$, the general prediction flow of Transformer-based deep tabular model is:

$$\text{Transformer}(X) = \text{Prediction}(\text{TransformerLayer}(\dots(\text{TransformerLayer}(X)))),$$
$$\text{Prediction}(X) = \text{Linear}(\text{ReLU}(\text{mean}(X))),$$

where the Transformer layer is implemented as

$$\text{TransformerLayer}(\boldsymbol{X}) = \text{LayerNorm}(\text{Residual}(\text{FFN}, \text{Residual}(\text{MultiheadAttention}, \boldsymbol{X}))),$$
$$\text{Residual}(\text{Module}, \boldsymbol{X}) = \boldsymbol{X} + \text{Dropout}(\text{Module}(\boldsymbol{X})),$$
$$\text{FFN}(\boldsymbol{X}) = \text{Linear}(\text{Dropout}(\text{ReGLU}(\text{Linear}(\boldsymbol{X})))),$$

where $\text{LayerNorm}(\cdot)$ is a normalization layer that performs layer normalization on the input (Ba et al., 2016). It computes the mean and standard deviation along specified axes and normalizes the tensor using these statistics. It is an alternative to $\text{BatchNorm}(\cdot)$ and is commonly used in Transformer where batch statistics are inappropriate. $\text{MultiheadAttention}(\cdot)$ is a key component of Transformer-based architectures, allowing the model to attend to multiple parts of the input sequence simultaneously and capture diverse patterns and relationships. $\text{ReGLU}(\cdot)$ is a GLU variant designed for Transformer (Shazeer, 2020).

## APPENDIX D    DATA

In this section, we introduce the detailed properties of source datasets and describe how to construct the synthetic datasets. We also introduce how to split real-world datasets into heterogeneous pre-training and fine-tuning datasets.

### D.1    DATA PREPROCESSING

For the sake of fair comparison, almost identical preprocessing steps are applied to all methods. In the case of categorical datasets, missing values are treated as a new discrete choice within each feature. As for numerical datasets, missing values are filled with the mean value of each respective feature. Standardization, involving mean subtraction and scaling, is applied to normalize each numerical dataset. To handle categorical features, encoding methods are employed to assist baselines that lack direct support. In the case of TabPFN (Hollmann et al., 2023), an ordinal encoder is utilized to ensure a limited quantity of encoded features for the method to work well on gpu. For other baselines, one-hot encoding is employed when necessary.

### D.2    SYNTHETIC DATASETS

We aim to construct a synthetic four-class classification dataset with semantically similar features and random features. As described in the main text, the dataset consists of six features $\{x_{:,i}\}_{i=1}^{6}$, each feature with four categorical choices. When constructing each sample in the dataset, the feature value is as follows ($p$ is the probability) :

- $x_{:,1} \in \{A, B, C, D\}$, $p = 0.25$ is assigned to each choice.
- $x_{:,2} \in \{E, F, G, H\}$, $p = 0.25$ is assigned to each choice.
- $x_{:,3} \in \{A', B', C', D'\}$, $x_{:,3} = x_{:,1}$ with $p = 0.8$; random choices otherwise.
- $x_{:,4} \in \{E', F', G', H'\}$, $x_{:,4} = x_{:,2}$ with $p = 0.8$; random choices otherwise.
- $x_{:,5}$ and $x_{:,6}$ are comletely random choices.

The label $y \in \{1, 2, 3, 4\}$ of a sample is assigned following the rules:

- when $x_{:,1} \in \{C, D\}$ and $x_{:,2} \in \{E, F\}$, $y = 1$ with $p = 0.8$; random choices otherwise.
- when $x_{:,1} \in \{A, B\}$ and $x_{:,2} \in \{G, H\}$, $y = 2$ with $p = 0.8$; random choices otherwise.
- when $x_{:,1} \in \{A, B\}$ and $x_{:,2} \in \{E, F\}$, $y = 3$ with $p = 0.8$; random choices otherwise.
- when $x_{:,1} \in \{C, D\}$ and $x_{:,2} \in \{G, H\}$, $y = 4$ with $p = 0.8$; random choices otherwise.

It is clearly shown in the rules of construction that, within 80% probability, $x_{:,1}$ and $x_{:,3}$, $x_{:,2}$ and $x_{:,4}$ have a one-to-one correspondence with the categorical choices, they are semantically similar features, while $x_{:,5}$ and $x_{:,6}$ are random noise features. We expect the feature tokens of $A'$ to be close to $A$, $B'$ close to $B$, *etc.* Besides, we aim at identifying the random features $x_{:,5}$ and $x_{:,6}$ from the feature tokens. As demonstrated in the main text, TABTOKEN incorporates these semantics into the feature tokens, fulfilling our aforementioned objectives.

Table 3: Descriptions of full datasets. There are different feature types and task types. These full datasets will be divided into pre-training datasets and fine-tuning datasets.

| Properties | Eye | Colon | Clave | Cardio | Jannis | Htru | Breast | Elevators | Super | Volume |
|---|---|---|---|---|---|---|---|---|---|---|
| Feature type | num | cat | cat | num | num | num | cat | num | num | num |
| Feature num | 26 | 18 | 16 | 11 | 54 | 8 | 29 | 18 | 81 | 53 |
| Task type | multiclass | binary | multiclass | binary | multiclass | binary | binary | regression | regression | regression |
| Size of full | 10.9K | 3.0K | 10.8K | 70.0K | 83.7K | 17.9K | 3.9K | 16.6K | 21.3K | 50.9K |
| Size of train | 7.0K | 1.8K | 6.9K | 44.8K | 53.6K | 11.5K | 2.3K | 10.6K | 13.6K | 32.8K |
| Size of val | 1.8K | 0.6K | 1.7K | 11.2K | 13.4K | 2.9K | 0.8K | 2.7K | 3.4K | 8.1K |
| Size of test | 2.2K | 0.6K | 2.2K | 14.0K | 16.7K | 3.6K | 0.8K | 3.3K | 4.3K | 10.0K |
| Source | OpenML | Datasphere | UCI | Kaggle | AutoML | UCI | Datasphere | OpenML | UCI | UCI |

Table 4: Descriptions of transferring datasets and fine-tuning datasets. We maintain an overlap of approximately 50% between the overlapping features and the fine-tuning features, which corresponds to the "medium" level among different overlapping ratios. "# Pre-training feature" denotes the number of pre-training features.

| Properties | Eye | Colon | Clave | Cardio | Jannis | Htru | Breast | Elevators | Super | Volume |
|---|---|---|---|---|---|---|---|---|---|---|
| # Pre-training feature | 17 | 13 | 11 | 7 | 36 | 6 | 20 | 12 | 54 | 35 |
| # Fine-tuning feature | 17 | 12 | 10 | 7 | 36 | 5 | 19 | 12 | 54 | 35 |
| # Overlapping feature | 8 | 7 | 5 | 3 | 38 | 3 | 10 | 6 | 27 | 17 |

### D.3 REAL-WORLD DATASETS AND EVALUATION SETUPS

To get datasets in different domains, the real-world datasets are from OpenML (Vanschoren et al., 2014), UCI, AutoML, Kaggle, and Projectdatasphere. The descriptions are shown in Table 3.

We randomly sample 20% instances to construct the test set. The remaining 80% instances are used for training. In the training set, we randomly hold out 20% of instances as the validation set. We split the whole training set into two parts. The first part consists of 80% of instances, which are utilized as pre-training dataset. The remaining instances are used for sampling few-shot downstream datasets. Expect for standard tabular tasks, the validation set is only used in the pre-training stage for saving the best checkpoint as the pre-training model.

To obtain a clearer explanation, consider a tabular dataset with nine features: f1 to f9. The upstream dataset comprises features f1 to f6, while the downstream dataset includes features f4 to f9.

The details of transferring dataset are shown in Table 4.

(1) **The number of shots is the sample size for each class**. For instance, a 5-shot dataset for binary classification consists of $5 \times 2$ samples. (2) **The overlapping ratio is defined as the ratio between the number of overlapping features and the total number of features used for fine-tuning**. The low, medium, and high overlapping ratios are shown in Table 5. In the main text, we investigate the cases with different number of shots or different overlapping ratios.

### APPENDIX E  ADDITIONAL ANALYSES AND EXPERIMENTS

We analyze TABTOKEN from various aspects. First, we investigate several possible implementations of "**Contrastive Token Regularization (CTR)**", and demonstrate that the simple contrastive form in the main text works the best. Then in subsection E.3, we test the ability of TABTOKEN when the target data share the same feature space but different distributions with the pre-trained model. We also explore the variant of TABTOKEN when the target data have entirely different feature and label spaces with the pre-trained model.

Table 5: Different overlapping ratios of transferring datasets. (l: low, m: medium, h: high). The overlapping ratios are calculated by dividing "# Overlapping feature" by "# Fine-tuning feature".

| Properties | Jannis (l) | Jannis (m) | Jannis (h) | Eye (l) | Eye (m) | Eye (h) |
|---|---|---|---|---|---|---|
| # Pre-training feature | 34 | 36 | 38 | 16 | 17 | 19 |
| # Fine-tuning feature | 34 | 36 | 38 | 15 | 17 | 18 |
| # Overlapping feature | 14 | 18 | 22 | 5 | 8 | 11 |
| Overlapping ratio (%) | 41 | 50 | 69 | 33 | 47 | 61 |

Table 6: We investigate the ability of TABTOKEN in standard tasks. We train deep models on the full pre-training data. TABTOKEN improves different deep models with either numerical or categorical features. (↑ ~ accuracy).

| | Eye↑ | Colon↑ | Clave↑ | Cardio↑ | Jannis↑ | Htru↑ | Breast↑ | Elevators↓ | Super↓ | Volume↓ |
|---|---|---|---|---|---|---|---|---|---|---|
| MLP | 0.4633 | 0.6299 | 0.7345 | 0.7317 | 0.5210 | 0.9801 | 0.8855 | 0.00191 | 13.4827 | 106.2745 |
| + CTR | 0.4849 | 0.6222 | 0.7412 | 0.7333 | 0.5373 | 0.9823 | 0.8772 | 0.00193 | 13.1038 | 105.2636 |
| ResNet | 0.4670 | 0.6206 | 0.7310 | 0.7318 | 0.5124 | 0.9779 | 0.8771 | 0.00192 | 13.8263 | 106.1255 |
| + CTR | 0.4725 | 0.6222 | 0.7326 | 0.7340 | 0.5195 | 0.9803 | 0.8783 | 0.00192 | 13.9194 | 105.8477 |
| Trans. | 0.4638 | 0.6414 | 0.7375 | 0.7325 | 0.5052 | 0.9807 | 0.8686 | 0.00191 | 13.4189 | 104.9273 |
| + CTR | 0.4698 | 0.6448 | 0.7467 | 0.7366 | 0.5215 | 0.9813 | 0.8804 | 0.00190 | 13.3295 | 104.0182 |

### E.1 TOKEN MATTERS IN IMPROVING DEEP MODELS

We investigate whether the token-based deep model can be improved through CTR. We directly evaluate the deep models in the full pre-training dataset. We use Optuna (Akiba et al., 2019) library and search hyperparameters for 30 iterations on the validation set. Table 6 shows the mean results over 10 random seeds. Different deep model architectures can achieve better prediction performance by coupling with TABTOKEN. TABTOKEN enhances the discriminative ability of deep models in standard tabular tasks. The results validate the potential of TABTOKEN beyond transfer tasks.

### E.2 ADDITIONAL ANALYSES AND COMPARISONS

**Visualization of feature tokens with vanilla training on bank-marketing dataset**. As shown in Figure 9, withour our CTR, the distribution of feature tokens is random.

**Different Forms of Contrastive Token Regularization**. In CTR, we aim to incorporate the semantics of features into tokens. One main intuition is that the tokens with similar feature semantics should be close while those tokens corresponding to different features may be far away from each other. Given a target dataset $\mathcal{D}^t = \{(\boldsymbol{x}_i^t, y_i^t)\}_{i=1}^{N^t}$ with $N^t$ examples and $d_t$ features, the feature tokenizer $h$ and feature combination (averaging or concatenating) process instance $\boldsymbol{x}_i^t$ to instance token $\boldsymbol{T}_i$. Assume there are $C$ classes in total, we denote the class center belonging to the target label $y_i$ of instance $\boldsymbol{x}_i^t$ as $\boldsymbol{S}_{y_i}$, while the class centers of different labels as $\boldsymbol{S}_{j \neq y_i}$. Recall that $h$ is the feature tokenizer and $g$ is the top-layer model. The CTR is usually optimized with the following objective:

$$\min_{h,g} \sum_{i=1}^{N^t} \left[ \ell(g \circ h(\boldsymbol{x}_i^t), y_i^t) \right] + \beta \Omega \left( \{\boldsymbol{T}_i\}_{i=1}^N \right).$$

Here are possible implementations of the token regularization $\Omega$:

- Vanilla CTR: the implementation that we used in TABTOKEN, which minimizes the distance between an instance token with its target class center:

$$\Omega_{\text{TabToken}} \left( \{\boldsymbol{T}_i\}_{i=1}^N \right) = \frac{1}{N^t} \sum_{i=1}^{N^t} \|\boldsymbol{T}_i - \boldsymbol{S}_{y_i}\|_2^2.$$

Table 7: The whole results in Table 1. †: TabPFN does not support regression tasks. ( ↑ ~ accuracy, ↓ ~ RMSE).

| | Eye↑ | Colon↑ | Clave↑ | Cardio↑ | Jannis↑ | Htru↑ | Breast↑ | Elevators↓ | Super↓ | Volume↓ |
|---|---|---|---|---|---|---|---|---|---|---|
| SVM | 0.3621 | 0.5921 | 0.3482 | 0.6036 | 0.3512 | 0.8376 | 0.8211 | 0.0088 | 33.9036 | 124.2391 |
| | ± 0.0044 | ± 0.0028 | ± 0.0094 | ± 0.0027 | ± 0.0048 | ± 0.0029 | ± 0.0049 | ± 0.0003 | ± 1.0362 | ± 1.3914 |
| XGBoost | 0.3699 | 0.5676 | 0.3506 | 0.5703 | 0.3222 | 0.8369 | **0.8453** | 0.0090 | 34.6605 | 123.9724 |
| | ± 0.0035 | ± 0.0074 | ± 0.0027 | ± 0.0047 | ± 0.0063 | ± 0.0028 | ± 0.0092 | ± 0.0002 | ± 1.7548 | ± 1.6470 |
| FT-trans | 0.3916 | 0.5792 | 0.3584 | 0.6064 | 0.3064 | 0.8252 | 0.8275 | 0.0081 | 31.3274 | 122.8319 |
| | ± 0.0075 | ± 0.0028 | ± 0.0083 | ± 0.0013 | ± 0.0047 | ± 0.0028 | ± 0.0018 | ± 0.0003 | ± 0.9462 | ± 1.0277 |
| TabPFN† | 0.3918 | 0.5809 | 0.3733 | 0.5965 | 0.3601 | 0.8371 | 0.7438 | - | - | - |
| | ± 0.0064 | ± 0.0082 | ± 0.0016 | ± 0.0037 | ± 0.0045 | ± 0.0027 | ± 0.0029 | | | |
| SCARF | 0.3371 | 0.6048 | 0.2144 | 0.5547 | 0.3523 | 0.8131 | 0.7063 | 0.0097 | 39.9343 | 124.5373 |
| | ± 0.0125 | ± 0.0152 | ± 0.0127 | ± 0.0094 | ± 0.0081 | ± 0.0058 | ± 0.0083 | ± 0.0004 | ± 1.4749 | ± 2.1749 |
| TabRet | 0.3477 | 0.4688 | 0.2393 | 0.4329 | 0.3545 | 0.8231 | 0.7252 | 0.0094 | 41.2537 | 126.4713 |
| | ± 0.0017 | ± 0.0048 | ± 0.0082 | ± 0.0038 | ± 0.0037 | ± 0.0037 | ± 0.0085 | ± 0.0002 | ± 1.4772 | ± 1,2734 |
| XTab | 0.3851 | 0.5964 | 0.3627 | 0.5856 | 0.2868 | 0.8363 | 0.8145 | 0.0077 | 38.5030 | 119.6656 |
| | ± 0.0085 | ± 0.0045 | ± 0.0074 | ± 0.0047 | ± 0.0092 | ± 0.0037 | ± 0.0019 | ± 0.0002 | ± 1.7463 | ± 0.8743 |
| ORCA | 0.3823 | 0.5876 | 0.3689 | 0.6042 | 0.3413 | 0.8421 | 0.8242 | 0.0082 | 37.9436 | 121.4952 |
| | ± 0.0037 | ± 0.0074 | ± 0.0018 | ± 0.0041 | ± 0.0082 | ± 0.0048 | ± 0.0054 | ± 0.0003 | ± 2.1852 | ± 1.2876 |
| TABTOKEN | **0.3982** | **0.6074** | **0.3741** | **0.6229** | **0.3687** | **0.8459** | 0.8284 | **0.0074** | **30.9636** | **118.7280** |
| | ± 0.0054 | ± 0.0061 | ± 0.0023 | ± 0.0024 | ± 0.0015 | ± 0.0034 | ± 0.0013 | ± 0.0002 | ± 1.5274 | ± 1.0285 |

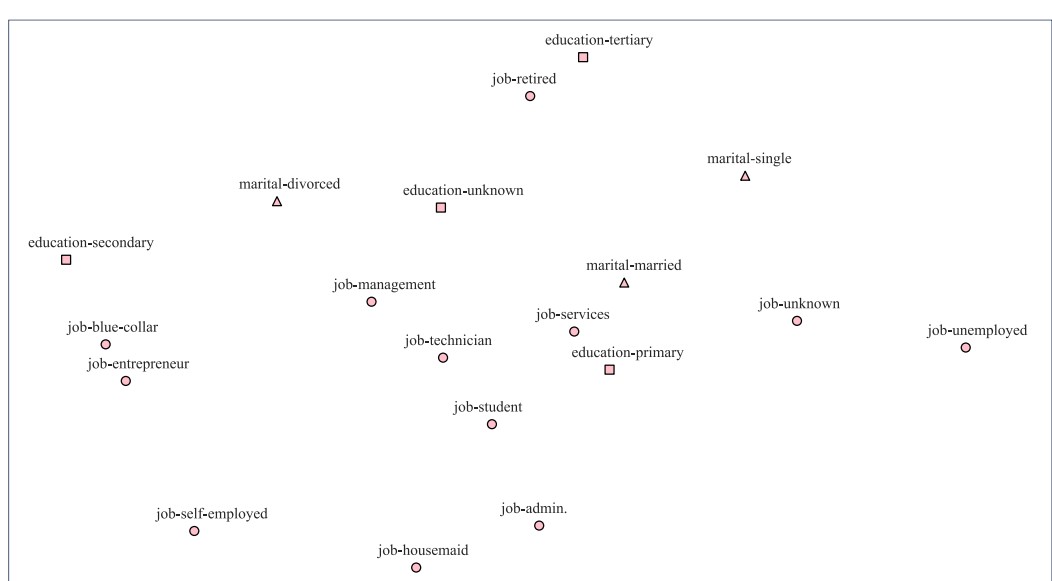

Figure 9: **Feature tokens with vanilla training on bank-marketing dataset.**

- Hardest: the objective is to push $T_i$ away from the nearest center of the different classes, which aims at moving instances away from the centers of the most easily misclassified labels.

$$\Omega_{\text{hardest}}\left(\{T_i\}_{i=1}^N\right) = \frac{1}{N^t}\sum_{i=1}^{N^t}\min(\{\|T_i - S_j\|_2^2\}_{j\neq y_i}).$$

- All-hard: the objective is to push $T_i$ away from all the centers of the different classes, which minimize the average distance between the instance and the centers of all other labels. (while in the binary classification task, "All-hard" and "Hardest" are the same).

$$\Omega_{\text{all}-\text{hard}}\left(\{T_i\}_{i=1}^N\right) = \frac{1}{N^t\cdot(C-1)}\sum_{i=1}^{N^t}\sum_{j\neq y_i}(\|T_i - S_j\|_2^2).$$

Table 8: Test accuracy on fine-tuning datasets for different forms of contrastive token regularization. We use TABTOKEN with different token regularization techniques on the pre-training dataset. We reuse the pre-trained feature tokenizer and apply token regularization during fine-tuning on the 5-shot downstream dataset. TABTOKEN is considered better overall, but in certain cases such as Jannis and Htru, further improvement can be achieved by incorporating another token regularization that focuses on pushing apart the centers of different classes. (↑ ~ accuracy, ↓ ~ RMSE).

| | Eye↑ | Colon↑ | Clave↑ | Cardio↑ | Jannis↑ | Htru↑ | Breast↑ | Elevators↓ | Super↓ | Volume↓ |
|---|---|---|---|---|---|---|---|---|---|---|
| $\Omega_{\mathrm{hardest}}$ | 0.3803 | 0.5895 | 0.3577 | 0.6155 | 0.3462 | 0.8422 | 0.8121 | 0.0083 | 35.8297 | 121.2847 |
| $\Omega_{\mathrm{all-hard}}$ | 0.3792 | 0.5895 | 0.3596 | 0.6155 | 0.3483 | 0.8422 | 0.8121 | 0.0083 | 35.8297 | 121.2847 |
| $\Omega_{\mathrm{supcon}}$ | 0.3904 | 0.5976 | 0.3701 | 0.6041 | 0.3204 | 0.8293 | 0.8233 | 0.0077 | 34.8366 | 119.9927 |
| $\Omega_{\mathrm{triplet}}$ | **0.3993** | 0.5983 | 0.3675 | 0.6055 | 0.3049 | 0.8431 | 0.8178 | 0.0082 | 33.1726 | 120.8255 |
| $\Omega_{\mathrm{TabToken}}$ | 0.3982 | **0.6074** | **0.3741** | **0.6229** | 0.3687 | 0.8459 | **0.8284** | **0.0074** | **30.9636** | **118.7280** |
| $\Omega_{\mathrm{TabToken+hard}}$ | 0.3911 | 0.6032 | 0.3678 | 0.6204 | **0.3936** | **0.8625** | 0.8253 | 0.0076 | 32.1836 | 118.7364 |

Table 9: Test accuracy of different methods on the 5-shot fine-tuning datasets with the help of pre-training. The construction of transferring datasets is shown in Table 4. Token loss allows tokens to gain potential for prediction, but the transferability of feature tokens is not as strong as in TABTOKEN. OPID, by utilizing model ensemble, has achieved better performance on Jannis and Htru. TABTOKEN shows superior performance on most datasets. (↑ ~ accuracy, ↓ ~ RMSE).

| | Eye↑ | Colon↑ | Clave↑ | Cardio↑ | Jannis↑ | Htru↑ | Breast↑ |
|---|---|---|---|---|---|---|---|
| Token head | 0.3863 | 0.6058 | 0.3683 | 0.6225 | 0.3675 | 0.8497 | 0.8059 |
| LR transfer | 0.3661 | 0.6001 | 0.3561 | 0.5947 | 0.3552 | 0.6834 | 0.7817 |
| OPID | 0.3903 | 0.5942 | 0.3689 | 0.6091 | **0.3754** | **0.8784** | 0.8143 |
| TABTOKEN | **0.3982** | **0.6074** | **0.3741** | **0.6229** | 0.3687 | 0.8459 | **0.8284** |

- Supcon: the supervised contrastive loss (Khosla et al., 2020). The objective remains the same, which is to bring instances of the same label closer together while keeping instances of different labels far apart. However, this approach requires more calculations based on the distances of instance tokens. We use the default configuration of official implementation.
- Triplet: the triplet contrastive loss with margin (Hermans et al., 2017). The objective is to ensure that the positive examples are closer to the anchor than the negative examples by at least the specified margin. We use the default configuration of official implementation.
- TABTOKEN + hard: the objective is the combination of CTR and All-hard:

$$\Omega_{\mathrm{TabToken+hard}} = \frac{1}{N^t} \sum_{i=1}^{N^t} \left( \|\boldsymbol{T}_i - \boldsymbol{S}_{y_i}\|_2^2 - \frac{1}{C-1} \sum_{j \neq y_i} \|\boldsymbol{T}_i - \boldsymbol{S}_j\|_2^2 \right).$$

We use TABTOKEN with different token regularization techniques. We reuse the pre-trained feature tokenizer and apply token regularization during fine-tuning on the 5-shot downstream dataset. The results in the Table 8 shows the test accuracy for downstream tasks with 5-shot. The same to the evaluation in the main text, for each method and dataset, we train on 30 randomly sampled few-shot subsets, reporting the performance averaged over 30 subsets and 10 random seeds. The only difference in training is the type of token regularization. Although $\Omega_{\mathrm{TabToken}}$ is the simplest objective, it allows the feature tokens of the pre-trained model to obtain better transferability.

**Different Methods for Feature Overlapping**. We compare baselines suitable for overlapping features, the descriptions are as follows:

- "Token loss" simultaneously train a linear classifier on the instance tokens during the training process of TABTOKEN and incorporating its loss into the final loss instead of CTR, the quality of feature tokens is expected to be improved by direct predicting based on the tokens.

Table 10: Test accuracy on the 5-shot fine-tuning datasets when the instance distributions are changed. (SD: same distribution, DD: different distributions). We add Gaussian noise to pre-training datasets to change the instance distributions. $\Delta$ is the average absolute change in prediction accuracy for the three methods when there is a distribution shift across five datasets. TABTOKEN can maintain its transferability and significantly outperform the other two transfer methods.

|  | Eye | Cardio | Jannis | Htru | $\Delta$ |
|---|---|---|---|---|---|
| SCARF (SD) | 0.3371 | 0.5547 | 0.3523 | 0.5938 | 0.0288 |
| SCARF (DD) | 0.3561 | 0.5320 | 0.3547 | 0.4958 | |
| TabRet(SD) | 0.3477 | 0.4329 | 0.3545 | 0.6305 | 0.0484 |
| TabRet (DD) | 0.3661 | 0.5063 | 0.3189 | 0.5175 | |
| TABTOKEN (SD) | 0.3982 | 0.6229 | 0.3678 | 0.8459 | **0.0037** |
| TABTOKEN (DD) | 0.3989 | 0.6189 | 0.3629 | 0.8479 | |

- In "LR transfer", the logistic regression classifier obtained from the pre-training set is directly used to initialize the fine-tuning classifier with the overlapping part. For features that are unseen in the fine-tuning phase, their corresponding weights are initialized to zero.
- In "OPID" (Hou & Zhou, 2018), during the pre-training phase, a sub-classifier is jointly trained on overlapping features, and the output of this sub-classifier is treated as knowledge from the pre-training set. This knowledge, in the form of new features, is concatenated with the fine-tuning dataset. During the fine-tuning phase, the sub-classifier and the new classifier are jointly trained, and the final prediction is the weighted sum of their predictions.

The results for these three baselines are in Table 9. Although OPID shows advantages in two datasets, when pre-training, it need the information about which features will be overlapping features in downstream task. While all features are treated equally during pre-training in TABTOKEN. Besides, OPID benefits from the ensemble of classifiers.

### E.3  EXTENSION TO MORE COMPLEX SCENARIOS

**Different Instance Distributions**. To explore the scenario where the feature space is the same, but the pre-training instance distribution differs from the fine-tuning distribution, we first split the full training set into two halves. In the first part, we construct the pre-training dataset by adding Gaussian noise. The standard deviation of noise is 10% of the feature's standard deviation. In the second part, we extract 30 5-shot sub-datasets as downstream tasks. To facilitate the addition of Gaussian noise, we conduct experiments on numerical datasets. The results are in Table 10. TABTOKEN is robust to the deviation of instance distributions, achieving the least changes in prediction performance.

**Different Feature and Label Space**. We explored transferring scenarios where the pre-training dataset and fine-tuning dataset are completely different. We use CTR to pre-train Transformer on Jannis, which owns a large number of features. We expect the downstream task to select feature tokens from these completely non-overlapping but semantically meaningful tokens using a re-weighting mechanism. By incorporating $n$ learnable new feature tokens $\{E_j^{\mathrm{new}}\}_{j=1}^n$ and a matching layer $W \in \mathbb{R}^{d_t \times (d+n)}$, we adapt TABTOKEN for scenarios with non-overlapping features.

We concatenate the new tokens with $d$ pre-trained feature tokens, constructing a "token library" $\left(\{E_j^{\mathrm{pre}}\}_{j=1}^d \cup \{E_j^{\mathrm{new}}\}_{j=1}^n\right) \in \mathbb{R}^{(d+n) \times k}$. The expression for re-weighting based TABTOKEN is as follows:

$$\{E_j^{\mathrm{fine}}\}_{j=1}^{d_t} = W \left(\{E_j^{\mathrm{pre}}\}_{j=1}^d \cup \{E_j^{\mathrm{new}}\}_{j=1}^n\right) \in \mathbb{R}^{d_t \times k}.$$

where $\{E_j^{\mathrm{fine}}\}_{j=1}^{d_t}$ is the feature tokens for fine-tuning feature tokenizer $h$, $\{E_j^{\mathrm{pre}}\}_{j=1}^d$ is the pre-trained feature tokens of $h_0$. $W$ is the re-weighting matrix for selecting feature tokens.

Overlapping feature transfer methods like TabRet may not be suitable for scenarios where there are non-overlapping features. TABTOKEN constructs a feature tokenizer by re-weighting the feature tokens in the pre-training set. We expect to obtain more useful feature tokens by training fewer parameters. Table 11 reports the performance with different number of fine-tuning features. Despite

Table 11: Test accuracy for different feature and label space. We conduct experiments on 5-shot Eye datasets with different feature numbers (shown in Table 5). We pre-train feature tokenizer with Transformer on full Jannis, fine-tuning on 20 randomly sampled few-shot subsets. re-weighting based TABTOKEN can benefit from pre-trained feature tokens, especially when the number of fine-tuning features is not large.

|  | Eye (l) | Eye (m) | Eye (h) |
|---|---|---|---|
| XGBoost | 0.3719 | 0.3699 | 0.3909 |
| CatBoost | 0.3759 | 0.3791 | 0.3974 |
| FT-trans | 0.3857 | 0.3916 | 0.3922 |
| TabPFN | 0.3885 | 0.3918 | **0.4081** |
| TABTOKEN | **0.3923** | **0.3977** | 0.3974 |

Table 12: Test accuracy for different tuning modules. We pre-train Transformer on pre-training datasets in Table 4. When we fix certain part of the pre-trained Transformer and fine-tune, the pre-training model have lower transferring effect than TABTOKEN. The frozen pre-trained top-layer model is not suitable for overlapping transfer scenarios. Feature tokens with semantics should be used for transfer. The best choice is to directly fine-tune the entire top-layer model as TABTOKEN.

|  | Eye | Cardio |
|---|---|---|
| Tune last layer | 0.3907 | 0.6155 |
| Tune attention | 0.3894 | 0.6217 |
| Tune linear | 0.3886 | 0.6153 |
| Fix top-layer | 0.3770 | 0.5983 |
| TABTOKEN | **0.3982** | **0.6229** |

the disparity between the pre-training and the fine-tuning dataset, re-weighting is a token search-like mechanism to enable the target task to benefit from the heterogeneous pre-trained feature tokens.

**Summary**. Among various contrastive token regularizations, simple CTR has demonstrated superior performance in transferring tasks. Compared to other methods, TABTOKEN is capable of obtaining easily transferable feature tokens during the pre-training phase. TABTOKEN maintains its transferability even when there are differences in instance distributions. Re-weighting based TABTOKEN proves to be effective when there is a non-overlapping feature set.

### E.4 ABLATION STUDY

We first adjust the modules tuned during the transfer process. Then, we conduct an ablation study based on different pre-training data sizes, model size, and token dimensions. Besides, We explore the combination of TABTOKEN and self-supervised loss.

**Tuning Modules**. In TABTOKEN, we do not fix the top-layer model when fine-tuning, while tuning all the Transformer, which indicates that feature tokens matter. We compare various tuning choices. When we train the fine-tuning tokenizer using TABTOKEN, we tune the last $\text{TransformerLayer}(\cdot)$, tune the $\text{MultiheadAttention}(\cdot)$, and tune the $\text{Linear}(\cdot)$ in Transformer layer while keeping other modules in Transformer frozen (the final prediction linear head is trainable). Besides, we conduct experiments on tuning the entire fine-tuning tokenizer with fixed top-layer model. The results in Table 12 show that token matters in transferring, we need to tune the entire top-layer model.

**Different Model Size**. We conducted an ablation study with different model sizes on four datasets, altering the number of layers in the transformer during pre-training. We keep the fine-tuning model's number of layers fixed at 3. We use the same setting as the experiments in Table 1. The experimental results in Table 13 indicate that when the pre-trained model size is relatively small (the number of layers is less than 3), the effectiveness of transfer is impacted. When the number of layers is three or more, the transfer capability of the tokenizer is challenging to further enhance. The impact of model

Table 13: Test accuracy for different model size. We use different number of layers for pre-training Transformer.

|  | Eye | Jannis | Cardio | Htru |
|---|---|---|---|---|
| layer num = 1 | 38.66 | 36.20 | 62.26 | 84.45 |
| layer num = 2 | 38.56 | 36.45 | 62.06 | 84.55 |
| layer num = 3 | 39.28 | **36.87** | 62.29 | 84.59 |
| layer num = 4 | 39.11 | 36.70 | **63.23** | **84.80** |
| layer num = 5 | **39.59** | 36.49 | 63.08 | 84.33 |

Table 14: Test accuracy for self-supervised token regularization. We pre-train feature tokenizer with self-supervised loss instead of CTR.

|  | Eye | Jannis | Cardio | Htru |
|---|---|---|---|---|
| cr = 0.1 | 38.10 | 31.46 | **62.40** | 82.35 |
| cr = 0.2 | 38.35 | 34.11 | 62.07 | 83.30 |
| cr = 0.3 | 38.22 | 35.59 | 61.84 | 83.55 |
| CTR | **39.28** | **36.87** | 62.29 | **84.59** |

size may be dependent on the difficulty of the task. More complex tasks require larger model sizes to achieve sufficient transferability.

**Self-supervised Pre-training without Annotations**. For each mini-batch of examples from the unlabeled training data, we generate a corrupted version for each example. We uniformly sample some fraction (corrupted ratio) of the features and replace each of those features with a random draw from that feature's empirical marginal distribution. We use normal distribution for numerical features. We then pass both example and corrupted example through the tokenizer and average their respective outputs. Finally, we L2-normalize the outputs and compute the InfoNCE loss instead of our CTR. We set the corruption ratio (cr) to 0.1, 0.2, and 0.3. Experimental results in Table 14 indicate that TABTOKEN with this self-supervised loss struggles to confer transferability to the model on the majority of datasets. We use the same setting as the experiments in Table 1.

**Tuning Token Dimension and Pre-training Size**. In our other experiments, we use a default token dimension of 64. Now, we will conduct experiments with token dimensions of 16, 32, 64, 128, and 256. Conventional transfer methods typically require a large amount of pre-training data. In our study, we randomly sample 20%, 40%, 60%, 80%, and 100% of the pre-training dataset to investigate the impact of data volume on the transfer performance of TABTOKEN. The results in Table 15 show that the larger dimension of tokens is not always better. TABTOKEN exhibits stable transfer performance even when the data volume decreases. TABTOKEN does not need a large amount of pre-training data to achieve the transfer effect.

**Complex Datasets**. We conducted experiments on two larger and more complex datasets. We collect scene recognition dataset (300 features, binary classification) and sylva agnostic dataset (217 features, binary classification) from OpenML. We use the same setting as the experiments in Table 1. The experimental results in Table 16 indicate that TABTOKEN continues to be an advantageous solution among deep models on complex datasets. XGBoost remains competitive on complex datasets.

## APPENDIX F    IMPLEMENTATION

In this section, we present the experimental configurations employed for the baselines and TABTOKEN. Given the absence of a validation dataset in the downstream few-shot task, we adopt default configurations to ensure a fair comparison. For standard tabular task, we follow the hyper-parameter space in (Gorishniy et al., 2021). All hyper-parameters are selected by Optuna library[1] with Bayesian

---

[1]https://optuna.org/

Table 15: Test accuracy for different token dimension and pre-training size. The transferring task is the same to Table 12. **Left**: The influence of token dimension. We change the dimension of feature tokens in tokenizer. We find that increasing the dimension of tokens does not lead to better transfer effect. **Right**: The transferring results with different size of pre-training dataset. We sample subsets from the pre-training dataset based on different proportions. TABTOKEN does not need a large amount of pre-training data to achieve the transfer effect.

|  | Eye | Cardio |
|---|---|---|
| 16 | 0.3909 | 0.5859 |
| 32 | **0.3999** | **0.6258** |
| 64 | 0.3982 | 0.6229 |
| 128 | 0.3977 | 0.6182 |
| 256 | 0.3949 | 0.6060 |

|  | Eye | Cardio |
|---|---|---|
| 20% | 0.3836 | 0.6205 |
| 40% | 0.3878 | 0.6207 |
| 60% | **0.3992** | 0.6179 |
| 80% | 0.3962 | 0.6185 |
| 100% | 0.3982 | **0.6229** |

Table 16: Test accuracy for complex datasets with more than 200 features. Due to GPU memory constraints, we set token dimension to 16 and configure the batch size as 256.

|  | XGBoost | XTab | TabToken |
|---|---|---|---|
| Scene | **56.24** | 42.97 | 53.87 |
| Sylva | 68.15 | 24.59 | **72.92** |

optimization over 30 trials. The best hyper-parameters are used and the average accuracy over 10 different random seeds is calculated.

**MLP and ResNet**. We use three-layer MLP and set dropout to 0.2. The feature token size is set to 64 for MLP, ResNet, and Transformer. The default configuration of ResNet is in the left of Table 17.

**Transformer**. The right tabular of Table 17 describes the configuration of FT-trans (Gorishniy et al., 2021) and Transformer layer in TABTOKEN.

**TabPFN**. We use the official implementation: https://github.com/automl/TabPFN and use the default configuration.

**SCARF and TabRet**. We use the default configutation in https://github.com/pfnet-research/tabret, which is also the official implementation of TabRet (Onishi et al., 2023). To ensure a fair comparison, we set the number of pre-training epochs to 200, patience to 20, fine-tuning epochs to 20. We modified the implementations to prevent the use of the validation set during the fine-tuning process.

**XTab**. We reuse the checkpoint with the highest number of training epochs from the official implementation of XTab (Zhu et al., 2023): https://github.com/BingzhaoZhu/XTab. We perform evaluations on the target datasets using XTab's light fine-tuning approach.

**ORCA**. We follow the configurations of ORCA (Shen et al., 2023) for OpenML11 datasets: text for embedder_dataset and roberta for pre-training model. (https://github.com/sjunhongshen/ORCA).

**TABTOKEN**. The configuration of feature tokenizer and top-layer models are described in Table 17. During pre-training, the learning rate is set to 1e-3, the batch size is set to 1024. During fine-tuning, the learning rate is set to 5e-4. We fine-tuning models in 10 epochs. We set $\beta = 1.0$ for both pre-training and fine-tuning.

Table 17: The parameters of ResNet and Transformer for top-layer models. **Left**: Default configuration used for ResNet. **Right**: Default configuration used for Tranformer which is also the configuration for implementing FT-trans.

| | |
|---|---|
| Layer count | 3 |
| Feature token size | 64 |
| Token bias | False |
| Layer size | 168 |
| Hidden factor | 2.9 |
| Hidden dropout | 0.5 |
| Residual dropout | 0.0 |
| Activation | ReLU |
| Normalization | BatchNorm |
| Optimizer | AdamW |
| Pre-train Learning rate | 1e-3 |
| Weight decay | 2e-4 |

| | |
|---|---|
| Layer count | 3 |
| Feature token size | 64 |
| Token bias | False |
| Head count | 8 |
| Activation & FFN size factor | ReGLU, $4/3$ |
| Attention dropout | 0.08 |
| FFN dropout | 0.3 |
| Residual dropout | 0.1 |
| Initialization | Kaiming |
| Optimizer | AdamW |
| Pre-train Learning rate | 1e-3 |
| Fine-Tune Learning rate | 5e-4 |
| Weight decay | 2e-4 |

