# OpenReview forum: "Unlocking the Transferability of Tokens in Deep Models for Tabular Data"
_ICLR.cc/2024/Conference — Submitted to ICLR 2024_

### Official Review · Reviewer_B5qv · 2023-10-29

**Soundness:** 2 fair
**Presentation:** 2 fair
**Contribution:** 2 fair
**Rating:** 5
**Confidence:** 3

**Summary:**

The authors propose TabToken which is a regularization & token feature learning method to incorporatee semantics at the token level during pre-training. This is specifically for tabular datasets with a mixture of real and categorical feature values. The authors claim that pretraining features and the top layer model during pre-training and subsequently finetuning the top layer allows better token feature transfers to other types of complex datasets.

**Strengths:**

- Consideration of real world problems: The authors considered using datasets that reflect the complexities of real world datasets where there is a mixture of data formats.
- Holistic testing: The authors evaluated their method on 10 tabular datasets spanning different domains.

**Weaknesses:**

- Significance: Lin et al. 2023 [1] demonstrated that the output layer of pre-trained text encoders do encode similarities or structure in tokens. This seems to contradict the authors claim and Figure 4 that vanilla training does not contain structure of the features? Is the lack of feature token transferability specific to tabular datasets? If so, is this solving a narrow or a domain specific problem, or would it generalize to any language model that uses a tokenizer?
- Incremental performance: Table 1 shows that TabToken only leads to a marginal improvement (< 0.2 accuracy and < 3 RMSE), some of the baselines are non deep learning methods. Figure 6, also shows marginal improvement of TabToken compared to other methods with increasing feature ratios. Table 2 also shows incremental improvement or even a decrease in performance when combining existing models with CTR. Is this why the improvement or deprovement with CTR was not highlighted using bold numbers in table 2?
- Difficulty of datasets used: The number of features in the datasets span between 8 to 54 in table 3. With a small dataset scale, non deep learning methods actually perform better. Could the paper be a case of using an over-engineered solution? More importantly, the benefits of using TabToken to improve deep learning algorithms becomes questionable since the dataset complexity is not at scale to employ deep learning methods. Perhaps pre-training or finetuning only the tokenizer might be relevant and the encoded features can be passed to a non deep learning classifier such as SVM? This could alleviate the complexities of training deep learning classifiers and focus the problem of learning good feature representations.

[1] Lin, Z., Azaman, H., Kumar, M. G., & Tan, C. (2023). Compositional Learning of Visually-Grounded Concepts Using Reinforcement. arXiv preprint arXiv:2309.04504.

**Questions:**

- will the algorithm be released as a pytorch or tensorflow module that can be incorporated into other models?
- what is the difference between TabToken and CTR? The difference can be delineated better.

**Details Of Ethics Concerns:**

The authors work is not of ethical concern. However, I am curious about the bank-marketing dataset that is accessible for ML researchers to train and build models on. I am wondering about its potential to include bias or unfair discrimination against the nature of jobs and the associated financial risk etc.

---

> ### Author Response · Authors · 2023-11-17
> **Respond to Reviewer B5qv (Part 1)**
>
> We express our gratitude to the reviewer for providing valuable feedback. Due to the character limit, we have divided our response into two parts.
>
> **Q1:** Pre-trained text encoders are able to encode similarities or structures in tokens. Is the lack of feature token transferability specific to tabular datasets?
>
> **A1:**
>
> BERT Text Encoder and CLIP Text Encoder, just as mentioned in the paper [1] cited by the reviewer, are pre-trained on vast corpora of data and have large-scale models. Through extensive data and pre-training methods for language model, the pre-trained text encoders possess semantic representation capabilities. However, due to the diverse feature types and different feature dimensions across different tabular tasks, pre-training for tabular data is typically conducted on a single task [A] or on multiple tasks without sharing tokenizers [B]. Because it is currently not feasible to simultaneously train a feature tokenizer on a large volume of heterogeneous tabular datasets, the pre-training tokenizer lacks token transferability.
>
> So the observation that the learned tabular tokens with poor semantics does not contradict with the semantic tokens of large NLP models. TabToken is solving a problem for tabular data where the learned tokens cannot encode similairities or structures, which makes the encoder of deep tabular models cannot be transfered as those in image processing and natural language processing domains.  In resource-constrained scenarios, our approach may generalize to a language model, assisting the text encoder in obtaining more semantic information.
>
> **Q2:** Some of the baselines are non-deep learning methods. The improvement is not enough.
>
> **A2:**
>
> Unlike the success observed in text and image datasets, deep models still struggle to outperform non-deep models on tabular data. Tree-based models remain competitive on tabular data even without accounting for their superior speed [C]. There is still no universally superior solution among GBDT and deep models [D]. Besides, in the experiments for XTab [B], with hyperparameter optimization, XTab ranks lower than XGBoost. Therefore, our comparison is between the currently strong baselines in both non-deep and deep methods. In our setting, TabToken ranks high across the majority of datasets. In Table 2, due to the diversity of datasets, it is challenging for CTR to achieve universal improvements. However, we still achieved improvements on the majority of datasets.
>
> One of the main contributions of our approach lies in enhancing the transferability of deep models. As shown in Table 12, when we transfer the pre-trained Transformer and fine-tune it, the pre-training model demonstrates a significantly lower transfer effect compared to TabToken. Therefore, we have made improvements in the transferability of deep tabular models.
>
> **Q3:** Non-deep learning methods perform better on small scale datasets. The dataset complexity is not at scale.
>
> **A3:**
>
> Firstly, the inferior performance of deep learning compared to non-deep models is not primarily due to small-scale issues. In comparisons on large-size datasets, this paper [C] found that, although the advantage of tree models diminishes a little, a gap still exists between deep models and tree models. (The Jannis dataset in our experiments falls into the category of large-size datasets as classified in this paper.) The challenges that hinder deep models from outperforming non-deep models on tabular data stem from various factors [C]: neural networks are inclined towards overly smooth solutions, MLP-like neural networks are more affected by uninformative features, and data lack invariance to rotation.
>
> Secondly, applying deep learning methods to small-scale tabular datasets has received attention in recent literature, such as TabPFN [E], where the deep learning methods are expected to possess the model prior and be transferred to downstream tasks more effectively. In our experiments, we show our TabToken can perform well on small-scale datasets, better than TabPFN.
>
> Thanks for the suggestions. We conducted experiments on two larger and more complex datasets. We collect scene recognition dataset (300 features, binary classification) and sylva agnostic dataset (217 features, binary classification) from OpenML. We use the same setting as the experiments in Table 1. The experimental results indicate that non-deep methods remain competitive on complex datasets, showing significant advantages in scene recognition dataset. Moreover, TabToken continues to be an advantageous solution among deep models on complex datasets.
>
> | Acc (%) | XGBoost   | XTab  | TabRet | TabToken  |
> | ------- | --------- | ----- | ------ | --------- |
> | scene   | **56.24** | 42.97 | 39.58  | 53.87     |
> | sylva   | 68.15     | 24.59 | 35.74  | **72.92** |
>
> (In Part 2, we will continue with the response.)

---

> > ### Author Response · Authors · 2023-11-17
> > **Respond to Reviewer B5qv (Part 2)**
> >
> > **Q4:** The encoded features can be passed to a non-deep learning classifier. This could alleviate the complexities of training deep learning classifiers and focus the problem of learning good feature representations.
> >
> > **A4:**
> >
> > Although tokens act as a kind of feature representation across tasks in visual and textual domains, the tokens are difficult to be shared on tabular data since a token corresponds to a particular feature, which can not be applied if the feature does not exist in the downstream task. Therefore, feature tokens need to be optimized.
> >
> > In downstream tasks, Tabtoken utilizes feature tokens  of pre-trained directly, while for unseen features, we still rely on the assistance of the top-layer model to learn new feature tokens. Therefore, in both upstream and downstream tasks, it is challenging to use a non-deep learning classifier for tokenizers.
> >
> > **Q5:** Will the algorithm be released as a pytorch or tensorflow module that can be incorporated into other models?
> >
> > **A5:**
> >
> > TabToken implements the idea through regularizing the learned tokens. The corresponding transferability of the model could be improved. It helps token-based methods in general, especially those deep tabular models.
> >
> > **Q6:** What is the difference between TabToken and CTR?
> >
> > **A6:**
> >
> > CTR is a component of TabToken. Specifically, TabToken is a methodology that encompasses the entire process, including how to pre-train, how to transfer, and how to fine-tune. On the other hand, CTR is a regularization term incorporated during the training of the model for both pre-training and downstream tasks.
> >
> > **Q7:** Ethic concern about bank-marketing.
> >
> > **A7:**
> >
> > We understand the concerns in this regard, and it is crucial to ensure non-discriminatory aspects. However, we need to clarify that the dataset does not have such issues. Bank-marketing is a classic task in the tabular data community, widely used in the majority of papers on tabular data [C,D,E,F]. As mentioned in our paper, TabTransformer [F] has also undergone visualization on this dataset. Additionally, it is a dataset that has been publicly available on UCI from 2012 (https://archive.ics.uci.edu/dataset/222/bank+marketing). The target is to predict whether the client will subscribe to a term deposit, instead of "financial risk". In conclusion, we believe there is no bias or unfair discrimination present in the dataset.
> >
> > [A] Revisiting Pretraining Objectives for Tabular Deep Learning. ArXiv, 2022.
> >
> > [B] Xtab: Cross-table pretraining for tabular transformers. ICML, 2023.
> >
> > [C] Why do tree-based models still outperform deep learning on tabular data? NeurIPS, 2022.
> >
> > [D] Revisiting Deep Learning Models for Tabular Data. NeurIPS, 2021.
> >
> > [E] TabPFN: A Transformer That Solves Small Tabular Classification Problems in a Second. ICLR, 2023.
> >
> > [F] TabTransformer: Tabular Data Modeling Using Contextual Embeddings. ArXiv, 2020.
> >
> > Thanks for the valuable suggestions. Please confirm if we have addressed the reviewer's concerns comprehensively. We are committed to promptly and thoroughly addressing any further questions.

---

> ### Comment · Area_Chair_Gv7M · 2023-11-20
>
> Hello, reviewer. Please review the author's response to see whether it addresses your concerns.

---

> > ### Comment · Reviewer_B5qv · 2023-11-21
> >
> > I thank the authors for your detailed rebuttal and running some additional experiments. I have revised my score to 5. I would recommend the authors including examples to distinguish tabular data types from other canonical data structures for clarity. The paper could potentially be an interesting discussion for ICLR. However, I am still struggling to understand the contribution of the work, while the application to real world data is appealing.

---

> > > ### Author Response · Authors · 2023-11-22
> > > **Respond to Reviewer B5qv**
> > >
> > > Thanks a lot for the reviewer's score improvement, and we are glad that our previous responses have addressed the concerns. In the following, we will provide a detailed explanation of the distinctions between tabular data and other canonical data structures. Additionally, we will elucidate the contributions of TabToken.
> > > - For textual data: Despite variations in specific tasks, different NLP tasks typically share similar vocabularies, allowing pre-training of tokenizers based on extensive data. By capturing long-term dependencies and syntactic structures in the text, word tokens in the tokenizer can acquire sufficient semantic knowledge, demonstrating transferability.
> > > - For image data: While images from different datasets may vary in content, pre-trained models can recognize and leverage universal image representations. Although the feature extractor in the image domain does not have a one-to-one correspondence like a tokenizer's lookup table, the extracted representation includes space structure and general characteristics of the image, exhibiting transferability.
> > >
> > > For tabular data: Tabular data is a highly structured form of data where each row represents an instance, and each column represents a different feature. Analogous to the one-word-one-token paradigm in textual tokenizers, token-based deep models for tabular data associate each feature with specific tokens.
> > >
> > > Just as there is a shared vocabulary in the NLP domain, there are also shared features in tabular data, such as 'bmi' and 'smoking status' in the health domain or 'job type' and 'education' in the financial domain. Downstream tasks may introduce new unseen features due to new task stages or objectives. Traditional tabular data models with fixed input dimensions from upstream tasks cannot be directly applied to downstream tasks. Besides, it is challenging to train efficient models on data-scarce downstream tasks. Therefore, we aim for transferability in tabular tokenizers, similar to NLP.
> > >
> > > However, due to distinct feature spaces across different tabular tasks, pre-training for tabular data is typically task-specific or conducted on multiple tasks without shared tokenizers. Therefore, conventional methods for tabular data transfer focus on upper-level (top-layer) models, such as transformers, requiring entirely different feature tokenizers for each task. Our observations in Section 5.1 indicate that feature tokenizers, due to the limited diversity in pre-training data and the absence of spatial dependencies for tabular tasks, fail to capture semantic information. Additionally, as analyzed in Appendix B, feature tokenizers do not enhance model capacity. Our proposed method enables learning semantics for feature tokens during pre-training and transferring them to downstream tasks. To the best of our knowledge, our approach is the first to emphasize the transferability of tokens on tabular data.
> > >
> > > The reviewer's feedback is crucial for enhancing the quality of our paper, and we will clarify the questions in the paper. Please let us know if this resolves the reviewer's concerns, and we are ready to provide further detailed responses. If our replies contribute to the understanding of the paper, we kindly request the reviewer to consider a further score adjustment. Thanks very much!

---

### Official Review · Reviewer_yFnT · 2023-10-30

**Soundness:** 3 good
**Presentation:** 3 good
**Contribution:** 2 fair
**Rating:** 6
**Confidence:** 3

**Summary:**

This paper proposes introducing semantics to feature tokens in order to improve the transferability of feature tokenizers through (1) averaging to represent an instance, (2) introducing a contrastive token regularization objective in pre-training to minimize the distance between instances and their respective class centers. Proposed solution is quite simple, but yet effective based on the empirical evaluation. I recommend the paper for acceptance, particularly given the importance of the problem they are solving in real-world applications.

**Strengths:**

(1) Clear and intuitive presentation.
(2) Well-motivated problem based on the analysis they conduct (see Figure 4 in particular).
(3) Simple but effective solution based on empirical analysis.

**Weaknesses:**

(1) They do not characterize the heterogeneity of the datasets they evaluate on -- hence, I do not have a sense for how hard it actually is to transfer between feature sets.
(2) There is no investigation on how the size of the models affect the described behavior of transferrability of feature tokenizers.

**Questions:**

n/a

---

> ### Author Response · Authors · 2023-11-17
> **Respond to Reviewer yFnT**
>
> We acknowledge and appreciate the reviewer's thoughtful comments. We respond to the concerns below:
>
> **Q1:** Characterize the heterogeneity of the datasets.
>
> **A1:**
>
> The heterogeneity of the dataset is reflected in the variations of the feature space. Specifically, there are features in the pre-training dataset that are absent in the downstream tasks, and vice versa, downstream tasks include features that were not encountered during the pre-training phase. The heterogeneity of the dataset can also be reflected in the distinct domains across datasets, as exemplified by the pre-training dataset BRFSS and the downstream datasets Diabetes & Stroke in Table 9. We construct a pre-training task on BRFSS for predicting HIV test experience. The task of Diabetes dataset is to predict whether a subject has diabetes. The Stroke dataset records clinical events and the task is to predict whether a subject is likely to get a stroke.
>
> **Q2:** How hard it actually is to transfer between feature sets.
>
> **A2:**
>
> Traditional tabular models, such as XGBoost, face challenges in adapting to changes in feature inputs during transfer. Each tabular feature has a strong correspondence with model parameters, making it hard to directly transfer pre-trained deep learning models when encountering unseen features. As demonstrated in our ablation study in Table 12, when we fix a certain part of the pre-trained Transformer and fine-tune it, the pre-training model exhibits a lower transfer effect compared to TabToken. The pre-trained deep model faces challenges in achieving the desired transfer effect between feature sets. The scarcity of available samples for fine-tuning on new datasets further complicates the knowledge transfer process.
>
> **Q3:** How the size of the models affect the described behavior of transferrability of feature tokenizers.
>
> **A3:**
>
> Following the reviewer's suggestion, we conducted an ablation study with different model sizes on four datasets, altering the number of layers in the transformer during pre-training. We keep the fine-tuning model's number of layers fixed at 3.   We use the same setting as the experiments in Table 1. The experimental results indicate that when the pre-trained model size is relatively small (the number of layers is less than 3), the effectiveness of transfer is impacted. When the number of layers is three or more, the transfer capability of the tokenizer is challenging to further enhance. The impact of model size may be dependent on the difficulty of the task. More complex tasks require larger model sizes to achieve sufficient transferability. The experimental results are as follows:
>
> | Acc (%)       | eye       | jannis    | cardio    | htru      |
> | ------------- | --------- | --------- | --------- | --------- |
> | layer num = 1 | 38.66     | 36.20     | 62.26     | 84.45     |
> | layer num = 2 | 38.56     | 36.45     | 62.06     | 84.55     |
> | layer num = 3 | 39.28     | **36.87** | 62.29     | 84.59     |
> | layer num = 4 | 39.11     | 36.70     | **63.23** | **84.80** |
> | layer num = 5 | **39.59** | 36.49     | 63.08     | 84.33     |
>
> Thanks for the suggestion. We will include this ablation study in the appendix.
>
>
>
> We thank for the beneficial commentary. Please let us know if our response has addressed the concerns. We're keen on promptly addressing any further comments from the reviewer.

---

> ### Comment · Area_Chair_Gv7M · 2023-11-20
>
> Hello, reviewer. The authors provide more information about the dataset and model size. Please review the author's response to see whether it addresses your concerns.

---

> > ### Comment · Reviewer_yFnT · 2023-11-21
> >
> > Thanks for responding to my questions and the additional ablation. My score remains the same.

---

> ### Author Response · Authors · 2023-11-21
> **Respond to Reviewer yFnT**
>
> Thanks a lot for the dedication and feedback. We are pleased that our responses addressed the questions. We have incorporated experiments on different model sizes as suggested by the reviewer into the revision (Table 13). If the reviewer has any other concerns or feedback, we look forward to engaging in more discussions and further enhancing the quality of our paper.

---

### Official Review · Reviewer_6daM · 2023-11-08

**Soundness:** 3 good
**Presentation:** 3 good
**Contribution:** 2 fair
**Rating:** 6
**Confidence:** 3

**Summary:**

This work presents a method to enhance the transferability of deep models on tabular data. Tabular data contain unique column features with their categorical or numerical values, making it challenging to transfer knowledge from pretraining data to unseen downstream data. To address this issue, the authors employ a contrastive regularization objective to learn semantic tokens for each column feature. Few-shot downstream tasks can leverage these overlapping semantic tokens. Experimental results demonstrate that the proposed method outperforms previous approaches in few-shot classification and regression tasks.

**Strengths:**

- The paper is well-written and easily understandable.
- The proposed method is simple yet effective, delivering superior performance on each dataset compared to previous methods.
- The paper provides a comprehensive analysis and visualization of the experiments.
- The proposed method exhibits a slight improvement in standard tabular tasks, which is a noteworthy point.

**Weaknesses:**

- The contrastive loss with the label has limited novelty.
- The section on related works should be integrated into the main article, as it is difficult to discern the specific improvements in comparison to previous methods.
- While the authors experimented with diverse domains of datasets, both the pretraining and finetuning datasets for each experiment originate from the same dataset. It remains uncertain whether the proposed method can be generalized across domains.
- The proposed method necessitates annotated labels for learning semantic tokens, limiting its application to supervised training. A self-supervised pretraining approach without annotations could be more appealing.

**Questions:**

Since the current model focuses on pretraining on one dataset and finetuning on the same dataset, is it feasible to explore pretraining on a large-scale cross-domain dataset?

---

> ### Author Response · Authors · 2023-11-17
> **Respond to Reviewer 6daM (Part 1)**
>
> We greatly appreciate the reviewer's valuable feedback. Due to the character limit, we divide our response into two parts.
>
> **Q1:** The contrastive loss with the label has limited novelty.
>
> **A1:**
>
> Previous transfer methods have focused on how to leverage large language models [A, B] or enhance the transferability of the top-layer model [C, D]. To the best of our knowledge, we are the first to focus on unlocking the transferability of feature tokens in the realm of tabular data transfer learning. We have unveiled a phenomenon starkly distinct from that observed in other domains: tokens learned by tabular data deep models lack semantic understanding. We devised the "Contrastive Token Regularization" (CTR), incorporating token pre-training and token fine-tuning tailored for tabular data.
>
> Although the general form of CTR is similar to the commonly used contrastive loss, the specific CTR is designed for deep tabular model. We start by converting a set of feature tokens to instance tokens through token combination. Then, we use token regularization to pull instance tokens towards their class center. This process enables contrastive loss to regularize the feature tokens effectively. Also, we have explored different forms of contrastive loss in Table 7.
>
> Moreover, we emphsize we can improve the semantic of the token through applying CTR over those tokens *before* the transformations in the deep tabular model, as we demonstrated in Figure 4 and Figure 5.
>
> Furthermore, CTR is just one component of TabToken, and the novelty of our approach also comes from the whole pre-training and fine-tuning method.
>
> **Q2:** Move the related work from the appendix to the main text.
>
> **A2:**
>
> Thanks for the suggestions.
>
> Although we discussed the relationship between our TabToken with previous methods such as Transferring Tabular Models across Feature Spaces, we chose to place the related work section at the beginning of the appendix due to the substantial content of the related work and constraints on the length of the main text.
>
> We will move the related work to the main text.
>
> **Q3:** Explore pre-training on a large-scale cross-domain dataset. Whether the proposed method can be generalized across domains.
>
> **A3:**
>
> In the appendix, we have explored the OOD generlization ability of TabToken in Table 9, where TabToken was pre-trained on BRFSS and fine-tuned on Diabetes & Stroke. BRFSS is the Behavioral Risk Factor Surveillance System dataset with more than 340K samples and more than 300 features from Kaggle. We construct a pre-training task on BRFSS for predicting HIV test experience. The task of Diabetes dataset is to predict whether a subject has diabetes. The Stroke dataset records clinical events and the task is to predict whether a subject is likely to get a stroke.
>
> Although these three datasets belong to different domains, they all fall under the category of health diagnosis. Approximately half of the features in the downstream datasets are present in the pre-training dataset, allowing us to apply TabToken. We conducted pre-training in the domain of health behaviours and transferred knowledge to domains for Diabetes or Stroke. The results (as shown in Table 9) indicate that TabToken outperforms other baselines in this natural cross-domain application with overlapping features. TabToken can be generalized across domains.
>
> Due to concerns of data privacy and security in healthcare and financial applications, access to lots of cross-domain datasets with overlapping features is typically restricted. We follow the dataset splitting in [E,F,G,H,I] to construct pre-training and fine-tuning tasks, where the resources are limited and it is challenging to obtain all features at once.
>
> [A] Tabllm: Few-shot classification of tabular data with large language models. AISTATS, 2023.
>
> [B] Transtab: Learning transferable tabular transformers across tables. NeurIPS, 2022.
>
> [C] Xtab: Cross-table pretraining for tabular transformers. ICML, 2023.
>
> [D] TabRet: Pre-training Transformer-based Tabular Models for Unseen Columns. ICLR workshop, 2023.
>
> [E] Heterogeneous ensemble for feature drifts in data streams. PAKDD, 2012.
>
> [F] One-pass learning with incremental and decremental features. IEEE T-PAMI, 2018.
>
> [G] Online Learning from Data Streams with Varying Feature Spaces. AAAI, 2019.
>
> [H] Learning with feature and distribution evolvablestreams. ICML, 2020.
>
> [I] Heterogeneous few-shot model rectification with semantic mapping. IEEE T-PAMI, 2021.
>
> (In Part 2, we will continue with the response.)

---

> > ### Author Response · Authors · 2023-11-17
> > **Respond to Reviewer 6daM (Part 2)**
> >
> > **Q4:** A self-supervised pre-training approach without annotations could be more appealing.
> >
> > **A4:**
> >
> > Firstly, during the upstream task, TabToken needs to train a predictive model. Based on this model, we complete the supervised upstream task while simultaneously enhancing model's transferability. It is different from pre-training a module specifically for the downstream task. Both the pre-training and downstream tasks can leverage our pre-trained model. SCARF [J], a baseline in the main text, is based on a self-supervised pre-training method. In our experimental results, SCARF struggles to demonstrate the benefits of self-supervision.
> >
> > Secondly, the relationship between features and labels is a crucial source of semantics. In response to the reviewer's suggestion, we follow SCARF to adopt a self-supervised approach. For each mini-batch of examples from the unlabeled training data, we generate a corrupted version $x'$ for each example $x$. We uniformly sample some fraction (corrupted ratio) of the features and replace each of those features with a random draw from that feature’s empirical marginal distribution. (Normal distribution for numerical features, probabilities for each categorical feature) . We then pass both $x'$ and $x$ through the tokenizer and average their respective outputs. Finally, we L2-normalize the outputs and compute the InfoNCE loss instead of our CTR. We set the corruption ratio (cr) to 0.1, 0.2, and 0.3. Experimental results on four datasets indicate that TabToken with this self-supervised loss struggles to confer transferability to the model on the majority of datasets. We use the same setting as the experiments in Table 1. The results are as follows:
> >
> > | Acc (%)                       | eye       | jannis    | cardio    | htru      |
> > | ----------------------------- | --------- | --------- | --------- | --------- |
> > | SCARF                         | 33.71     | 35.27     | 55.47     | 81.31     |
> > | TabToken (self-sup, cr = 0.1) | 38.10     | 31.46     | **62.40** | 82.35     |
> > | TabToken (self-sup, cr = 0.2) | 38.35     | 34.11     | 62.07     | 83.30     |
> > | TabToken (self-sup, cr = 0.3) | 38.22     | 35.59     | 61.84     | 83.55     |
> > | TabToken (CTR)                | **39.28** | **36.87** | 62.29     | **84.59** |
> >
> > Thanks for the suggestion. We will include the relevant experiments in our paper.
> >
> >
> >
> > [J] Scarf: Self-supervised contrastive learning using random feature corruption. ICLR, 2022.
> >
> >
> >
> > We appreciate the valuable feedback once more! Please feel free to inform us if our response has effectively addressed the concerns. We remain enthusiastic about promptly attending to any further questions or comments the reviewer may have.

---

> > > ### Comment · Reviewer_6daM · 2023-11-20
> > > **Response after author rebuttal**
> > >
> > > Thanks for the explanation and extra experiments. The authors clarify most of my questions. The result in Appendix Table 9 is good to highlight for cross-dataset transferability, showing it in the main text can further strengthen the contribution. I raised the score to 6.

---

> > > > ### Author Response · Authors · 2023-11-21
> > > > **Respond to Reviewer 6daM**
> > > >
> > > > We greatly appreciate the score improvements from the reviewer. We have incorporated the valuable suggestions from the reviewer in the updated revision,  discussing related work on the main text, introducing experiments focusing on unsupervised loss (Table 14), and relocating the cross-domain experiments to the main text (Table 2).  Furthermore, we have incorporated more experiments in the appendix, specifically investigating the impact of different pre-trained model sizes and evaluating TabToken on highly complex datasets.
> > > >
> > > > Thanks for the valuable suggestions to imporve the quality of our paper, we would be grateful if the reviewer could potentially further raise the score based on the improvements implemented. If the reviewer has any further concerns and suggestions, please do not hesitate to raise them. We will respond promptly and address them accordingly.

---

> ### Comment · Area_Chair_Gv7M · 2023-11-20
>
> Hello, reviewer. Please review the author's response. Does the response address your concern about the novelty?

---

### Author Response · Authors · 2023-11-21
**For all reviewers: paper revision**

We express our gratitude to the AC for the diligent efforts in facilitating communication. Thanks to all reviewers for their assistance in improving the quality of our paper. We have updated the revision based on the reviewers' valuable feedback.

- Following Reviewer 6daM's suggestions, we discuss related work on the main text (Section 2, detailed in appendix A), introduce experiments focusing on unsupervised loss (Table 14), and relocate the cross-domain experiments to the main text (Table 2).
- Following Reviewer yFnT's suggestions, we investigate the impact of different pre-trained model sizes (Table 13).
- Following Reviewer B5qv's suggestions, we evaluate TabToken on highly complex datasets (Table 16).

The Table index during the previous rebuttal process will be modified as follows: Table 2 $\rightarrow$ Table 6, Table 3 $\rightarrow$ Table 4, Table 7 $\rightarrow$ Table 8,  Table 9 $\rightarrow$ Table 2.

---

### Author Response · Authors · 2023-11-23
**Thanks to AC and Reviewers; Conclusion and future outlook**

We extend our heartfelt appreciation to the AC for facilitating our communication. Thanks a lot to Reviewer 6daM, yFnT, and B5qv, for their thorough evaluation of the paper and their perceptive and invaluable feedback. In this rebuttal, we have given careful thought to the reviewers’ suggestions and answered the questions and concerns.



We thank to all reviewers for identifying the highlight, affirming that the concept of "unlocking the semantic capturing ability and transferability of tokens in tabular deep models" is  **well-motivated based on our analysis** (Reviewer yFnT),  **simple yet effective** (Reviewer 6daM), and a **potentially interesting subject to discuss** (Reviewer B5qv). They consider our paper is **well-written and easily understandable** (Reviewer 6daM),  and is **presented clearly and intuitively** (Reviewer yFnT). Our paper **provides a comprehensive analysis and visualization of the experiments** (Reviewer 6daM).  Besides, solving **important** (Reviewer yFnT) and **challenging** (Reviewer 6daM) problem in real-world applications is **appealing** (Reviewer B5qv).



We have addressed the concerns raised by the reviewers and received recognition. We are very grateful for the constructive feedback and score improvement. In response to the valuable comments from the reviewers, **we have made revisions during the rebuttal stage. Due to the time constraints of the rebuttal, we will continue to refine the paper after the discussion.** We plan to incorporate more highly complex datasets as suggested (such as adding more datasets), further incorporate important discussions (such as additional clarifying the differences in token transferability between tabular data and other forms), focus more on cross-domain task (such as providing additional details on the datasets and experimental setups), ... We would like to emphasize that we have addressed these aspects in the rebuttal, and in the future, we aim to refine them further.



Thanks once again to the AC and reviewers for their dedication. We are excited about the opportunity to present a new paper revision!

---

### Meta-Review · Area_Chair_Gv7M · 2023-12-03

**Metareview:**

The paper presents a method for improving the transferability of deep models in dealing with tabular data. This is achieved through a contrastive regularization objective that aids in learning semantic tokens for each column feature.

The reviewers expressed concerns regarding the reliance on labelled data during pre-training and the generalizability of the method across domains. The authors have also made revisions based on the reviewers' suggestions, including introducing experiments focusing on unsupervised loss and evaluating the model on more complex datasets. A critical point raised by the reviewers pertains to the novelty of employing contrastive loss in this context. The claim that the contrastive loss is uniquely tailored for tabular data might be an overstatement.

**Justification For Why Not Higher Score:**

Reviewers expressed skepticism regarding the novelty of using contrastive loss, particularly challenging the claim that it is specifically designed for tabular data. This concern suggests that while the approach is effective, its novelty might not be as pronounced as claimed, impacting the potential for a higher score.

Reviewers pointed out the model's reliance on labeled data during pre-training and questioned its generalizability across different domains. While the authors have addressed these concerns to some extent, there remains room for improvement in demonstrating the model's versatility and effectiveness in a broader range of applications.

**Justification For Why Not Lower Score:**

N/A

---

### Decision · Program_Chairs · 2024-01-16

Reject